# Interpretability in the Wild: a Circuit for Indirect Object Identification in GPT-2 small

**Kevin Wang**[*], **Alexandre Variengien**[*], **Arthur Conmy**[*], **Buck Shlegeris**[†], **Jacob Steinhardt**[†‡§]
[†]Redwood Research
[‡]UC Berkeley

## Abstract

Research in mechanistic interpretability seeks to explain behaviors of ML models in terms of their internal components. However, most previous work either focuses on simple behaviors in small models, or describes complicated behaviors in larger models with broad strokes. In this work, we bridge this gap by presenting an explanation for how GPT-2 small performs a natural language task that requires logical reasoning: indirect object identification (IOI). Our explanation encompasses 28 attention heads grouped into 7 main classes, which we discovered using a combination of interpretability approaches including causal interventions and projections. To our knowledge, this investigation is the largest end-to-end attempt at reverse-engineering a natural behavior "in the wild" in a language model. We evaluate the reliability of our explanation using three quantitative criteria–*faithfulness, completeness* and *minimality*. Though these criteria support our explanation, they also point to remaining gaps in our understanding. Our work is a case study demonstrating a first step toward a better understanding of pre-trained language models, opening opportunities to scale to both larger models and more complex tasks.[1]

## 1    Introduction

Transformer-based language models (Vaswani et al., 2017; Brown et al., 2020) have demonstrated an impressive suite of capabilities, but largely remain black boxes. Understanding these models is difficult because they employ complex non-linear interactions in densely-connected layers and operate in a high-dimensional space. Despite this, they are already deployed in high-impact settings, underscoring the urgency of understanding and anticipating possible model behaviors. Some researchers have even argued that interpretability is necessary for the safe deployment of advanced machine learning systems (Hendrycks & Mazeika, 2022).

Work in mechanistic interpretability aims to discover, understand and verify the algorithms that model weights implement by reverse engineering model computation into human-understandable components (Olah, 2022; Meng et al., 2022; Geiger et al., 2021; Geva et al., 2020). By understanding underlying mechanisms, we can better predict out-of-distribution behavior (Mu & Andreas, 2020), identify and fix model errors (Hernandez et al., 2021; Vig et al., 2020), and understand emergent behavior (Nanda & Lieberum, 2022; Barak et al., 2022; Wei et al., 2022).

In this work, we aim to understand how GPT-2 small (Radford et al., 2019) implements a natural language task. To do so, we locate components of the network that produce specific behaviors, and study how they compose to complete the task. We do so by using *circuits analysis* (Räuker et al., 2022), identifying an induced subgraph of the model's computational graph that is human-understandable and responsible for completing the task. We employed a number of techniques, most notably activation patching, knockouts, and projections, which we believe are useful, general techniques for circuit discovery.[2]

---

[*]Work done while at Redwood Research. [§]Correspondence to `jsteinhardt@berkeley.edu`
[1]A full and up-to-date version of this work can be found at `https://arxiv.org/abs/2211.00593`
[2]We included an overview of the techniques used in Appendix L.

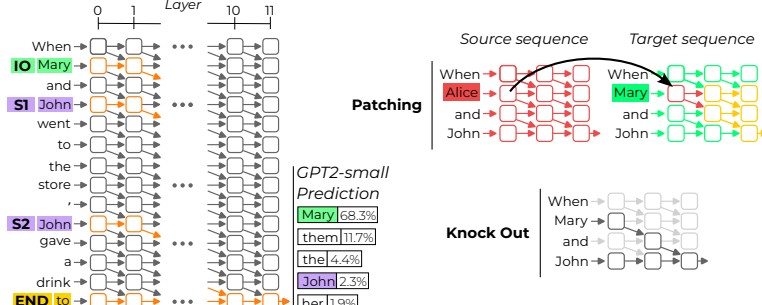

Figure 1: Left: We isolated a *circuit* (in orange) responsible for the flow of information connecting the indirect object 'Mary' to the next token prediction. The nodes are attention blocks and the edges represent the interactions between attention heads. Right: We discovered and validated this circuit using activation experiments, including both patches and knockouts of attention heads.

We focus on understanding a non-trivial, algorithmic natural language task that we call Indirect Object Identification (IOI). In IOI, sentences such as 'When Mary and John went to the store, John gave a drink to' should be completed with 'Mary'. We chose this task because it is linguistically meaningful and admits a complex but interpretable algorithm (Section 3).

We discover a circuit of 28 attention heads–1.5% of the total number of (head, token position) pairs–that completes this task. The circuit uses 7 different categories of heads (see Figure 2) to implement the algorithm. Together, these heads route information between different name tokens, to the end position, and finally to the output. Our work provides, to the best of our knowledge, the most detailed attempt at reverse-engineering a natural end-to-end behavior in a transformer-based language model.

Explanations for model behavior can easily be misleading or non-rigorous (Jain & Wallace, 2019; Bolukbasi et al., 2021). To remedy this problem, we formulate three criteria to help validate our circuit explanations. These criteria are **faithfulness** (the circuit can perform the task as well as the whole model), **completeness** (the circuit contains all the nodes used to perform the task), and **minimality** (the circuit doesn't contain nodes irrelevant to the task). Our circuit shows significant improvements compared to a naïve (but faithful) circuit, but fails to pass the most challenging tests.

In summary, our main contributions are: (1) We identify a large circuit in GPT-2 small that performs indirect-object identification on a specific distribution (Figure 2 and Section 3); (2) Through example, we identify useful techniques for understanding models, as well as surprising pitfalls; (3) We present criteria that ensure structural correspondence (in the computational graph abstraction) between the circuit and the model, and check experimentally whether our circuit meets this standard (Section 4).

## 2 BACKGROUND

In this section, we introduce the IOI task (an original contribution of this work), the transformer architecture, define *circuits* more formally and describe a technique for "knocking out" model nodes.

**Task description.** In indirect object identification (IOI), two names (the indirect object (IO) and the first occurrence of the subject (S1)) are introduced in an initial dependent clause (see Figure 1). A main clause then introduces the second occurrence of the subject (S2), who is usually exchanging an item. The task is to complete the main clause, which always ends with the token 'to', with the non-repeated name (IO). We create many dataset samples for IOI ($p_{IOI}$) using 15 templates (see Appendix A) with random single-token names, places and items.

We investigate the performance of GPT-2 small on this task. We study the original model from Radford et al. (2019), pretrained on a large corpus of internet text and without any fine-tuning. To quantify GPT-2 small performance on the IOI task, we used the *logit difference* between the logit values placed on the two names, where a positive score means the correct name (IO) has higher probability. This is also the difference in loss the model would receive in training if IO was correct compared to if S was correct. We report this metric averaged over $p_{IOI}$ throughout the paper. GPT-2 small has mean logit difference of 3.55 averaged across over 100,000 dataset examples.

**Transformer architecture.** GPT-2 small is a decoder-only transformer with 12 layers and 12 attention heads per attention layer. In this work, we mostly focus on understanding the mechanisms of attention heads, which we describe using notation similar to Elhage et al. (2021). We leave a full description of the model to Appendix E.

The input to the transformer is the sum of position and token embeddings, $x_0 \in \mathbb{R}^{N \times d}$, where $N$ is the number of tokens in the input and $d$ is the model dimension. This input embedding is the initial value of the *residual stream*, which all attention layers and MLPs read from and write to. Attention layer $i$ of the network takes as input $x_i \in \mathbb{R}^{N \times d}$, the value of the residual stream before it. The attention layer output can be decomposed into the sum of attention heads $h_{i,j}$. If the output of the attention layer is $y_i = \sum_j h_{i,j}(x_i)$, then the residual stream is updated to $x_i + y_i$.

Focusing on individual heads, each head $h_{i,j}$ is parametrized by four matrices $W_Q^{i,j}, W_K^{i,j}, W_V^{i,j} \in \mathbb{R}^{d \times \frac{d}{H}}$ and $W_O^{i,j} \in \mathbb{R}^{\frac{d}{H} \times d}$. We rewrite these parameters as low-rank matrices in $\mathbb{R}^{d \times d}$: $W_{OV}^{i,j} = W_O^{i,j} W_V^{i,j}$ and $W_{QK}^{i,j} = (W_Q^{i,j})^T W_K^{i,j}$. The QK matrix is used to compute the attention pattern $A_{i,j} \in \mathbb{R}^{N \times N}$ of head $(i, j)$, while the OV matrix determines what is written into the residual stream. At the end of the forward pass, a layer norm is applied before the unembed matrix $W_U$ projects the residual stream into logits.

## 2.1 CIRCUITS

In mechanistic interpretability, we want to reverse-engineer models into interpretable algorithms. A useful abstraction for this goal are *circuits*. If we think of a model as a computational graph $M$ where nodes are terms in its forward pass (neurons, attention heads, embeddings, etc.) and edges are the interactions between those terms (residual connections, attention, projections, etc.), a circuit $C$ is a subgraph of $M$ responsible for some behavior (such as completing the IOI task). This definition of a circuit is slightly different from that in Olah et al. (2020), where nodes are features (meaningful directions in the latent space of a model) instead of model components.

## 2.2 KNOCKOUTS

Just as the entire model $M$ defines a function $M(x)$ from inputs to logits, we also associate each circuit with a function $C(x)$, via *knockouts*. A knockout removes a set of nodes $K$ in a computational graph $M$ with the goal of "turning off" nodes in $K$ but capturing all other computations in $M$. Thus, $C(x)$ is defined by knocking out all nodes in $M \backslash C$ and taking the resulting logit outputs in the modified computational graph.

A first naïve knockout approach consists of simply deleting each node in $K$ from $M$. The net effect of this removal is to *zero ablate* $K$, meaning that we turn its output to 0. This naïve approach has an important limitation: 0 is an arbitrary value, and subsequent nodes might rely on the average activation value as an implicit bias term. Because of this, we find zero ablation to lead to noisy results in practice.

To address this, we instead knockout nodes through *mean ablation*: replacing them with their average activation value across some reference distribution (similar to the bias correction method used in Nanda & Lieberum (2022)). Mean-ablations will remove the influence of components sensitive to the *variation* in the reference distribution (i.e. attention heads that move names in $p_{\text{IOI}}$), but will not influence components using information *constant* in the distribution (i.e. attention patterns that are constant in $p_{\text{IOI}}$). Through mean-ablations, we are interested in finding the components that move information about names, which is the core of the IOI task and also varies with the distribution.

In this work, all knockouts are performed in a modified $p_{\text{IOI}}$ distribution with three random names, so the sentences no longer have a single plausible IO. We mean-ablate on this distribution, which we call the 'ABC' distribution, because mean-ablating on the $p_{\text{IOI}}$ distribution would not remove enough information, like information constant in $p_{\text{IOI}}$ that is helpful for the task. To knockout a single node, a (head, token position) pair in our circuit, we compute the mean of that node across samples of the same template. Computing means across the entire distribution instead of templates would average activations at different tokens, like names, verbs and conjunctions, mixing information destructively.

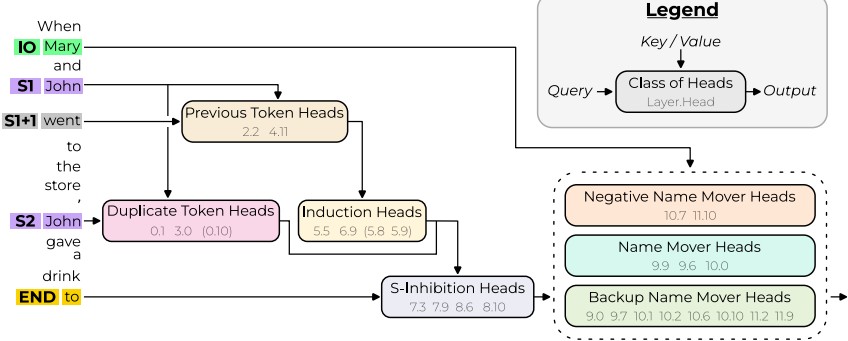

Figure 2: We discover a circuit in GPT-2 small that implements IOI. The input tokens on the left are passed into the residual stream. Attention heads move information between residual streams: the query and output arrows show which residual streams they write to, and the key/value arrows show which residual streams they read from.

## 3 DISCOVERING THE CIRCUIT

We seek to explain how GPT-2 small implements the IOI task (Section 2). Recall the example sentence "When Mary and John went to the store, John gave a drink to". We discovered that GPT-2's internal mechanisms implement the following human-interpretable algorithm to perform IOI:

1. Identify all previous names in the sentence (Mary, John, John).
2. Remove all names that are duplicates (in the example above: John).
3. Output the remaining name.

Our circuit contains three major classes of heads, corresponding to these three steps:

- *Duplicate Token Heads* identify tokens that have already appeared in the sentence. They are active at the S2 token, attend primarily to the S1 token and write a 'signal' into the residual stream that token duplication has occurred.
- *S-Inhibition Heads* perform step 2 of the human-interpretable algorithm. They are active at the END token, attend to the S2 token and write to bias the query of the Name Mover Heads against both S1 and S2 tokens.
- *Name Mover Heads*, by default, attend to previous names in the sentence, but due to the S-Inhibition Heads attend less to the S1 and S2 tokens. Their OV matrix is a name copying matrix, so in $p_{IOI}$, they increase the logit of the IO token.

A fourth major family of heads writes in the opposite direction of the Name Mover Heads, thus decreasing the confidence of the predictions. We speculate that these *Negative Name Mover Heads* might help the model "hedge" so as to avoid high cross-entropy loss when making mistakes.

There are also three minor classes of heads that perform related functions to the components above:

- *Previous Token Heads* copy the embedding of S to position S+1.
- *Induction Heads* perform the same role as the Duplicate Token Heads through an induction mechanism. They are active at position S2, attend to token S+1 (mediated by the Previous Token Heads), and output a signal that the S token previously appeared in the context.
- Finally, *Backup Name Mover Heads* do not normally move the IO token to the output, but take on this role if the regular Name Mover Heads are knocked out.

Note that our circuit does not include the MLPs. We are interested in the flow of information across tokens, and MLPs only process features along tokens. Moreover, initial investigations suggest all MLPs except for the first one are not crucial for this task (Appendix I), though more precise investigation is left for future work.

Below, we show step-by-step how we discovered each component, providing evidence that they behave as described above. We found that it was most natural to uncover the circuit starting at the logits and working back. Thus we start with the Name Mover and Negative Name Mover Heads.

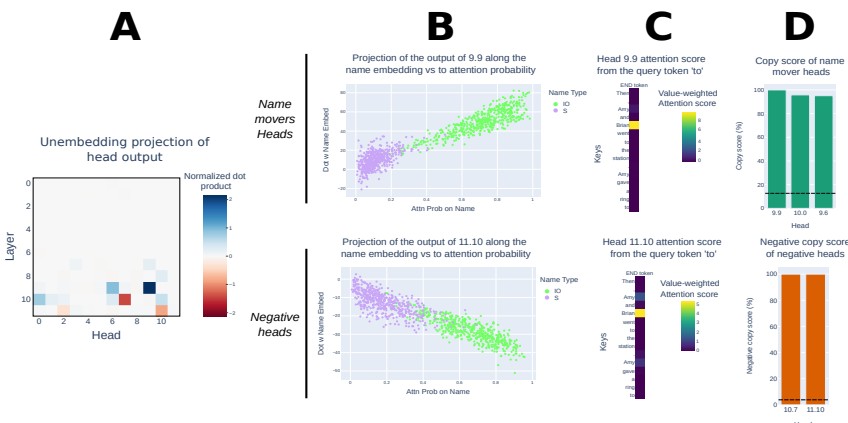

Figure 3: **A:** Name Movers and Negative Name Movers Heads are the heads that most strongly write in the $W_U[IO] - W_U[S]$ direction. **B**: Attention probability vs projection of the head output along $W_U[IO]$ or $W_U[S]$ respectively. Note that for S tokens, we sum the attention probability on both S1 and S2. **C**: Value-weighted attention score with the query at the end token. **D, top**: Positive copying score for the Name Mover Heads. **D, bottom**: Negative copying score for the Negative Name Mover Heads. Dashed lines are the average scores for all heads.

## 3.1 WHICH HEADS DIRECTLY WRITE TO THE OUTPUT? (NAME MOVER HEADS)

We begin by identifying which attention heads directly affect the model's output: in other words, the heads writing in the residual stream at the END position, in a direction that has high dot product with the logit difference. Formally, let $W_U$ denote the unembedding matrix, $\overline{\text{LN}}$ a layer norm operation (see Appendix H) and $W_U[IO]$, $W_U[S]$ the corresponding unembedding vectors for the $IO$ and $S$ tokens. We searched for heads $(i, j)$ such that

$$\lambda_{i,j} \overset{\text{def}}{=} \mathbb{E}_{X \sim p_{\text{IOI}}}[\langle \overline{\text{LN}} \circ h_{i,j}(X), W_U[IO] - W_U[S] \rangle]$$

had large magnitude. Recall that $h_{i,j}(X)$ is the value that head $(i, j)$ writes into the residual stream on input $X$. Therefore, heads with $\lambda_{i,j} > 0$ correctly promote the IO token over the S token (on average). The unembedding projection in (3.1) is called the *logit lens* and has been used in previous work to interpret intermediate activations (nostalgebraist, 2020) and parameters (Dar et al., 2022). We display the values of $\lambda_{i,j}$ in Figure 3 A. We see that only a few heads in the final layers have large logit projection $\lambda_{i,j}$. Specifically, 9.6, 9.9, and 10.0 have a large positive score, while 10.7 and 11.10 have a large negative score.

**Name Mover Heads.** To understand the positive heads, we first study their attention patterns. We find that they attend strongly to the IO token: the average attention probability of all heads over $p_{\text{IOI}}$ is 0.59. Since attention patterns can be misleading (Jain & Wallace, 2019), we check whether attention is correlated with the heads' functionality. We do so by scatter plotting the attention probability against the logit score $\langle h_i(X), W_U[IO] \rangle$. The results are shown in Figure 3 B: higher attention probability on the IO token is linearly correlated with higher output in the IO direction (correlation $\rho > 0.81$, $N = 500$). Based on this result, we hypothesize that these heads (i) attend to names and (ii) copy whatever they attend to. We therefore call these heads *Name Mover Heads*.

To check that the Name Mover Heads copy names generally, we studied what values are written via the heads' OV circuits. We transform the output of the first layer at a name token through the OV matrix of a Name Mover Head and then project to the logits. The copy score is the proportion of samples that contain the input name token in the top 5 logits ($N = 1000$). We find that all three Name Mover Heads have a copy score above 95% (compared to less than 20% for an average head).

**Negative Name Mover Heads.** In Figure 3, we also observed two heads strongly writing opposite the $W_U[IO] - W_U[S]$ direction. We called these heads *Negative Name Mover Heads*. Their copy score is calculated with the negative of their OV matrix. As described in Figure 3, they share all the properties of Name Mover Heads, except they write in the opposite of names they attend to.

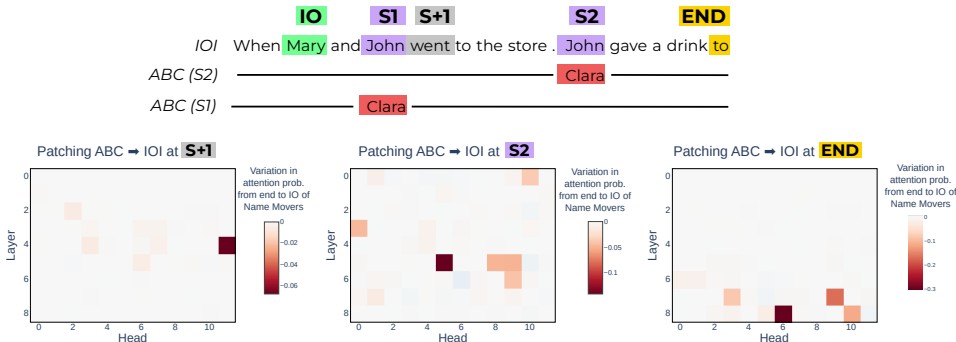

Figure 4: The attention probability to IO averaged over three Name Mover Heads is decreased most by the Previous Token Heads (left), Induction Heads (center) and S-Inhibition Heads (right) when we patch these attention heads from a sentence with a different S2 name (center and right), or a different S1 name (left).

## 3.2 WHICH HEADS AFFECT THE NAME MOVER HEADS' ATTENTION? (S-INHIBITION HEADS)

Given that the Name Mover Heads are primarily responsible for constructing the output, we ask why these Name Mover Heads pay preferential attention to the IO token. First, there are two ways to affect the Name Mover Heads's attention: through the query vector at the END token or the key vector at the IO token. Since the key vector appears early in the context, it likely does not contain much task-specific information, so we focus on the END query vector.

Then, by investigating Name Mover Heads on the ABC distribution (where the three names are distinct; see Section 2.2), we observed that their attention is not selective: they pay equal attention to the first two names. We thus ask: what has changed from the ABC distribution to the $p_{IOI}$ distribution to cause the Name Mover Heads to attend to the IO token preferentially?

To empirically answer this question, we perform a *patching* experiment, a similar type of causal intervention as performed in Meng et al. (2022); Vig et al. (2020). As illustrated in Figure 1 this technique consists of two steps. First we save all activations of the network run on a *source* sequence. Then we run the network on a *target* sequence, replacing some activations with the activations from the source sequence. We can then measure the behavior of the patched model. Doing this for each node individually locates the nodes that explain why model behavior is different in the source and target sequences.

In our case, we run activation patching with source sentences from the ABC distribution and target sentences from $p_{IOI}$. We then compute the change in attention probability from END to IO, averaged over the three Name Mover Heads. Since the Name Mover Heads attention on the IO is high in the $p_{IOI}$ distribution and low in ABC, patching at important heads from ABC to $p_{IOI}$ should decrease Name Mover Heads attention on IO. The results from patching every head at the END token position are shown in Figure 4, right. We observe that patching heads 7.3, 7.9, 8.6, 8.10 causes a decrease in the attention probability on IO, indicating that they are counterfactually important for the Name Mover Heads's attention probability on the IO token. We call these heads **S-Inhibition Heads**.

## 3.3 WHAT INFORMATION DO THE S-INHIBITION HEADS MOVE?

How do the S-Inhibition Heads differentiate between IO and S, so they inhibit one but not the other? We measured their attention pattern and found that they preferentially attend to the S2 token. We therefore studied what information these heads move from the S2 token position to the END position. We studied both the properties of the input, and which upstream affect the S-inhibition heads. Surprisingly, we found that the S-inhibition heads mostly depend on the repetition at the two positions where the S token occurs (Appendix G).

To study the heads that affect the S-inhibition heads, we ran a patching experiment at S2 from the ABC distribution to the IOI distribution and measured the variation in Name Mover Heads attention. The results (Figure 4, center) reveal a large set of heads influencing Name Mover Heads' attention

that did not appear at the END position. S-Inhibition Heads must mediate this effect, as they are the only heads influencing Name Mover Heads at the END position. This reasoning suggests that the outputs of this set of heads is moved by S-Inhibition Heads from S2 to the END token. When we analyze the attention patterns of these heads, we see two distinct groups emerge.

**Duplicate Token Heads.** One group attends from S2 to S1. We call these Duplicate Token Heads on the hypothesis that they detect duplicate tokens. To validate this, we analyze their attention pattern on sequences of random tokens (with no semantic meaning), we found that 2 of the 3 Duplicate Token Heads pay strong attention to a previous occurrence of the current token if it exists (see Appendix F for more details).

**Induction Heads and Previous Token Heads.** The other group of heads attends from S2 to S1+1 (the token after the S1 token): the classic attention pattern of an induction head. Previously described in Elhage et al. (2021), induction heads recognize the general pattern `[A] [B] ... [A]` and contribute to predicting `[B]` as the next token. For this, they act in pair with a Previous Token Head. The Previous Token Head should write information about `[A]` into the residual stream at `[B]`, so that the Induction Head can match the next occurrence of `[A]` to that position (and subsequently copy `[B]` to the output).

We therefore seek to identify Previous Token Heads used by our purported Induction Heads. To this end, we patched activations from a sentence where S1 is replaced by a random name, at the S+1 token index. As shown in figure 4, some heads (and particularly 4.11) appear to influence Name Mover Heads. Then, by looking at the attention pattern of the most important heads in this patching experiment, we identified 3 Previous Token Heads. We find that 2 of the 3 Previous Token Heads and 2 of the 4 Induction Heads demonstrated their expected attention patterns (Appendix F).

### 3.4 Did we miss anything? The Story of the Backup Name Movers Heads

Each type of head in our circuit has many copies, suggesting that the model implements redundant behavior. To make sure that we didn't miss any copies, we knocked out *all* of the Name Mover Heads at once. To our surprise, the circuit still worked (only 10% drop in logit difference). In addition, many heads write along $W_U[IO] - W_U[S]$ after the knockout, which did not do so previously.

We kept the heads with the largest $\lambda_{i,j}$, and call them *Backup Name Mover Heads*. See appendix B for further details on these heads. Among the height heads identified, we investigated their behavior before the knockout. We observe diverse behavior: 3 heads show close resemblance to Name Mover Heads; 3 heads equally attend to IO and S and copy them; 1 head pays more attention to S1 and copies it; 1 head seems to track and copy subjects of clauses, copying S2 in this case.

## 4 Experimental validation

In this section, we check that our circuit provides a good account of GPT-2's true behavior. In general, our introduced criteria depend on a measure $F$ of the performance of a circuit on a task. In our case, suppose $X \sim \mathsf{p}_{\text{IOI}}$, and $f(C(X); X)$ is the logit difference between the IO and S tokens when the circuit $C$ is run on the input $X$. The average logit difference $F(C) \stackrel{\text{def}}{=} \mathbb{E}_{X \sim \mathsf{p}_{\text{IOI}}}[f(C(X); X)]$ is a measure of how much a circuit predicts IO rather than S, i.e performs the IOI task.

Firstly, we check that $C$ is **faithful** to $M$, i.e. that it computes similar outputs. We do so by measuring $|F(M) - F(C)|$, and find that it is small: 0.2, or only 6% of $F(M) = 3.55$.

In Section 4.1 we define a running toy example of a model $M$ for which faithfulness is not sufficient to prescribe which circuits explain a behavior defined by a measure $F$ well. This motivates the criteria of completeness and minimality that we then check on our circuit. In addition to the criteria, we also validated our knowledge of the circuit by designing adversarial examples (see Appendix C).

### 4.1 Completeness

As a running example, suppose a model $M$ uses two similar and disjoint serial circuits (where each node depends on the previous node) $C_1$ and $C_2$. The two sub-circuits are run in parallel before applying an OR operation to their results. Identifying only one of the circuits is enough to achieve

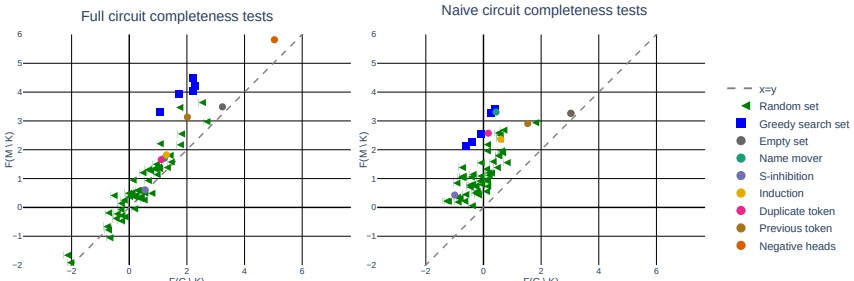

Figure 5: Plot of points $(x_K, y_K) = (\mathrm{F}(M \setminus K), \mathrm{F}(C \setminus K))$ for our circuit (left) and a naive circuit (right). Each point is for a different choice of $K$: 50 uniformly randomly chosen $K \subseteq C$, $K = \emptyset$, and the five $K$ with the highest incompleteness score found by greedy optimization. Since the incompleteness score is $|x_K - y_K|$, we show the line $y = x$ for reference.

faithfulness, but we want explanations that include both $C_1$ and $C_2$, since these are both used in the model.

To solve this problem, we introduce the **completeness** criterion: for every subset $K \subseteq C$, the *incompleteness score* $|F(C \setminus K) - F(M \setminus K)|$ should be small. In other words, $C$ and $M$ should not just be similar, but remain similar under knockouts.

In our running example, we can show that $C_1$ is not complete by setting $K = C_1$. Then $C_1 \setminus K$ is the empty circuit while $M \setminus K$ still contains $C_2$. The metric $|F(C_1 \setminus K) - F(M \setminus K)|$ will be large because $C_1 \setminus K$ has trivial performance while $M \setminus K$ successfully performs the task.

The criterion of completeness requires a search over exponentially many subsets $K \subseteq C$. This is computationally intractable given the size of our circuit, hence we use three sampling methods to find examples of $K$ that give large incompleteness score:

- The first sampling method chooses subsets $K \subseteq C$ uniformly at random.
- The second sampling method set $K$ to be an entire class of circuit heads $G$, e.g the Name Mover Heads. $C \setminus G$ should have low performance since it's missing a key component, whereas $M \setminus G$ might still do well if it has redundant components that fill in for $G$.
- Thirdly, we greedily optimized $K$ node-by-node to maximize the incompleteness score (see appendix K for the detail of the optimization procedure).

These first two methods of sampling $K$ suggested to us that our circuit was $\varepsilon$-complete for a small value of $\varepsilon$. However, the third resulted in sets $K$ that had high incompleteness score: up to 3.09. All such results are found in figure 5, on the left.

## 4.2 MINIMALITY

A faithful and complete circuit may contain unnecessary components, and so be overly complex. To avoid this, we should check that each of its nodes $v$ is necessary. This can be evaluated by knocking out a set of nodes $K$ and showing that adding back $v \in K$ to the circuit can significantly recover $F$.

Formally, the **minimality** require that for every node $v \in C$ there exists a subset $K \subseteq C \setminus \{v\}$ that has minimality score $|F(C \setminus (K \cup \{v\})) - F(C \setminus K)| \geq A$. We call such a circuit $A$-**minimal**.

In the running example, $C_1 \cup C_2$ is $A$-minimal for some non-trivial $A$. We can sketch a proof of this result given an informal definition of 'non-trivial'. To show this, note that if $v_1 \in C_1$ and $K = C_2$, then the minimality score is equal to $|F(C_1 \setminus \{v_1\}) - F(C_1)|$ which is large since $C_1$ is a serial circuit and so removing $v_1$ will destroy the behavior. We then proceed symmetrically for $v_2 \in C_2$.

In practice, we need to exhibit for every $v$ a set $K$ such that the minimality score is at least $A$. For most heads, removing the class of heads $G$ that $v$ is a part of provides a reasonable minimality score. We describe the sets $K$ that are required for them in Appendix J. The importance of individual nodes is highly variable, but they all have a significant impact on the final metric (at least 3% of the original logit difference). These results ensure that we did not interpret irrelevant nodes, but do show that the individual contribution of some single attention heads is small.

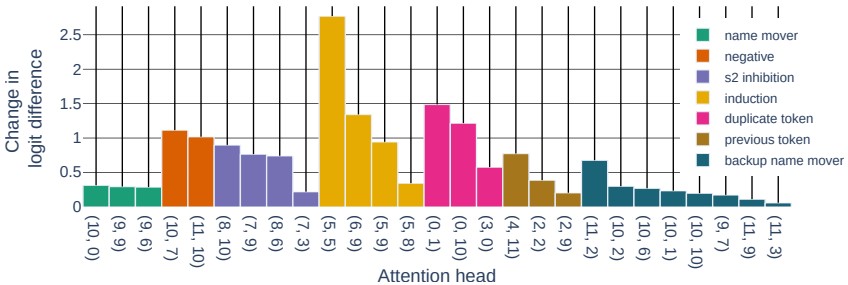

Figure 6: Plot of minimality scores $|F(C \setminus (K \cup \{v\})) - F(C \setminus K)|$ for all components $v$ in our circuit. The sets $K$ used for each component, as well as the initial and final values of the logit difference for each of these $v$ is in Appendix J. Our circuit is 0.06-minimal.

### 4.3 COMPARISON WITH A NAIVE CIRCUIT

In order to get a relative sense of the success of our explanation by our criteria, we compare the results on a naïve circuit that consists of the Name Mover Heads (but no Backup Name Mover Heads), S-Inhibition Heads, two Induction Heads, two Duplicate Token Heads and two Previous Token Heads. This circuit has a faithfulness score 0.1, a score comparable to our circuit's faithfulness score. However, contrary to our circuit, the naive circuit can be easily proven incomplete: by sampling random sets or by knocking-out by classes, we see that $F(M \setminus K)$ is much higher than $F(C \setminus K)$ (Figure 5, left). Nonetheless, when we applied the greedy heuristic to optimize for the incompleteness score, both circuits have similarly large incompleteness scores. Thus, we conclude that our worst-case completeness criterion was too high a bar, which future work could use as a high standard to validate circuit discovery.

## 5 DISCUSSION

In this work, we isolated, understood and validated a set of attention heads in GPT-2 small composed in a circuit that identifies indirect objects. Along the way, we discovered interesting structures emerging from the model internals that complicated the study. For instance, we identified heads compensating for the loss of function of other heads, and heads contributing negatively to the next-token prediction. Early results suggest that the latter phenomenon occurs for other tasks beyond IOI (see Appendix F).

However, our work also has several limitations. First, despite the detailed analysis presented here, we do not understand several components. Those include the attention patterns of the S-Inhibition Heads, and the effect of MLPs and layer norms. Second, the number of parameters in GPT-2 small is orders of magnitude away from state-of-the-art transformer language models. A future challenge is to scale this approach to these larger models. Thirdly, we only looked at the difference in average metric (logit difference) between the circuit and the model in order to compare how they both did the IOI task (Section 4). Looking at the average difference in metric between the circuit and model on individual examples would be a more stringent way to compare them, but it had too much variability to help us find a circuit. Fourthly, the definition of the task is limited: we only measure a fraction of the prediction made by the model, and do not study cases where the model is *not* performing IOI. Finally, more work is needed to validate the structural validation criterion we introduce here.

We hope that our work spurs further efforts in mechanistic explanations of larger language models computing different natural language tasks, with the eventual goal of understanding full language model capabilities.

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

## A  IOI TEMPLATES

We list all the template we used in Table 7. Each name was drawn from a list of 100 English first names, while the place and the object were chosen among a hand made list of 20 common names. All the word chosen were one token long to ensure proper sequence alignment computation of the mean activations.

| Templates in p_IOI |
|---|
| Then, [B] and [A] went to the [PLACE]. [B] gave a [OBJECT] to [A] |
| Then, [B] and [A] had a lot of fun at the [PLACE]. [B] gave a [OBJECT] to [A] |
| Then, [B] and [A] were working at the [PLACE]. [B] decided to give a [OBJECT] to [A] |
| Then, [B] and [A] were thinking about going to the [PLACE]. [B] wanted to give a [OBJECT] to [A] |
| Then, [B] and [A] had a long argument, and afterwards [B] said to [A] |
| After [B] and [A] went to the [PLACE], [B] gave a [OBJECT] to [A] |
| When [B] and [A] got a [OBJECT] at the [PLACE], [B] decided to give it to [A] |
| When [B] and [A] got a [OBJECT] at the [PLACE], [B] decided to give the [OBJECT] to [A] |
| While [B] and [A] were working at the [PLACE], [B] gave a [OBJECT] to [A] |
| While [B] and [A] were commuting to the [PLACE], [B] gave a [OBJECT] to [A] |
| After the lunch, [B] and [A] went to the [PLACE]. [B] gave a [OBJECT] to [A] |
| Afterwards, [B] and [A] went to the [PLACE]. [B] gave a [OBJECT] to [A] |
| Then, [B] and [A] had a long argument. Afterwards [B] said to [A] |
| The [PLACE] [B] and [A] went to had a [OBJECT]. [B] gave it to [A] |
| Friends [B] and [A] found a [OBJECT] at the [PLACE]. [B] gave it to [A] |

Figure 7: Templates used in the IOI dataset. All templates in the table fit the 'BABA' pattern, but we also use templates that fit the 'ABBA' pattern as well (not included for simplicity).

## B  BACKUP NAME MOVER HEADS

Here we discuss in more detail the discovery of the Backup Name Mover Heads. As shown in figure 8, knocking-out the three main Name Mover Heads doesn't leave the rest of the heads in a similar state as before. They seem to "compensate" the loss of function from the Name Mover Heads such that the logit difference is only 10% lower. We observe that the Negative Name Mover Heads head write less negatively in the direction of $W_U[IO] - W_U[S]$, 10.7 even write positively in this direction afterwards, while other heads that wrote slightly along $W_U[IO] - W_U[S]$ before the knock-out becomes the main contributor. Both the reason and the mechanism of this compensation effect are still unclear, we think that this could be an interesting phenomenon to investigate in future works. Among those last categories, we identify S-inhibition heads and a set of other head that we called *Backup Name Mover Heads*. We arbitrarily chose to keep the height heads that were not part of any other groups, and wrote in the direction of $W_U[IO] - W_U[S]$ above the threshold of 0.05.

In figure 9 we analyze the behavior of those newly identified heads with similar techniques as Name Mover Heads. Those can be grouped in 4 categories.

- 3 heads (10.1, 10.10 and 10.6) that behave similarly as Name Mover Heads according to their attention pattern, and scatter plots of attention vs dot product of their output with $W_U[IO] - W_U[S]$ (as 10.10).

- 3 heads (10.2, 11.9, 11.3) that pay equal attention to S1 and IO and wrote both of them (as 10.2 in Figure 9).

- One head, 11.2, that pays more attention to S1 and write preferentially in the direction of $W_U[S]$

- One head, 9.7, that pays attention to S2 and write negatively.

We did not thoroughly investigate this diversity of behavior, more work can be done to precisely describe these heads. However, these heads are also the ones with the less individual importance for the task (as shown by their minimality score in Figure 6). The exact choice of Backup Name Mover Heads doesn't change significantly the behavior of the circuit.

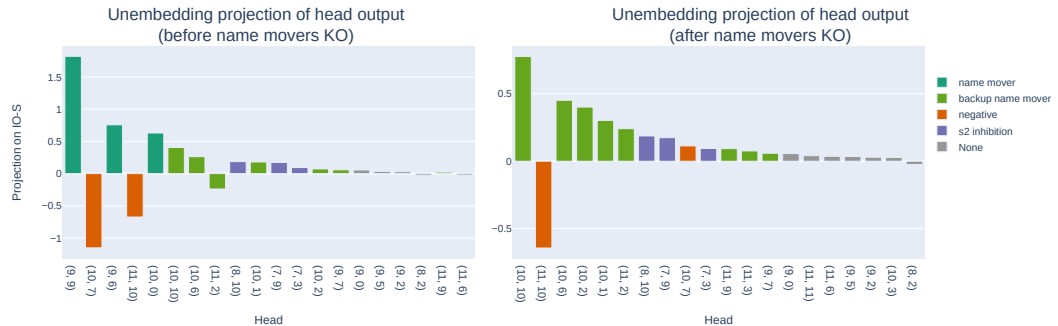

Figure 8: Discovery of the Backup Name Mover Heads. After knock-out of the Name Mover Heads (right) some heads write stronger in the $W_U[IO]$ or $W_U[S]$ direction than before (left). We also observed that negative heads seems inhibited by this operation.

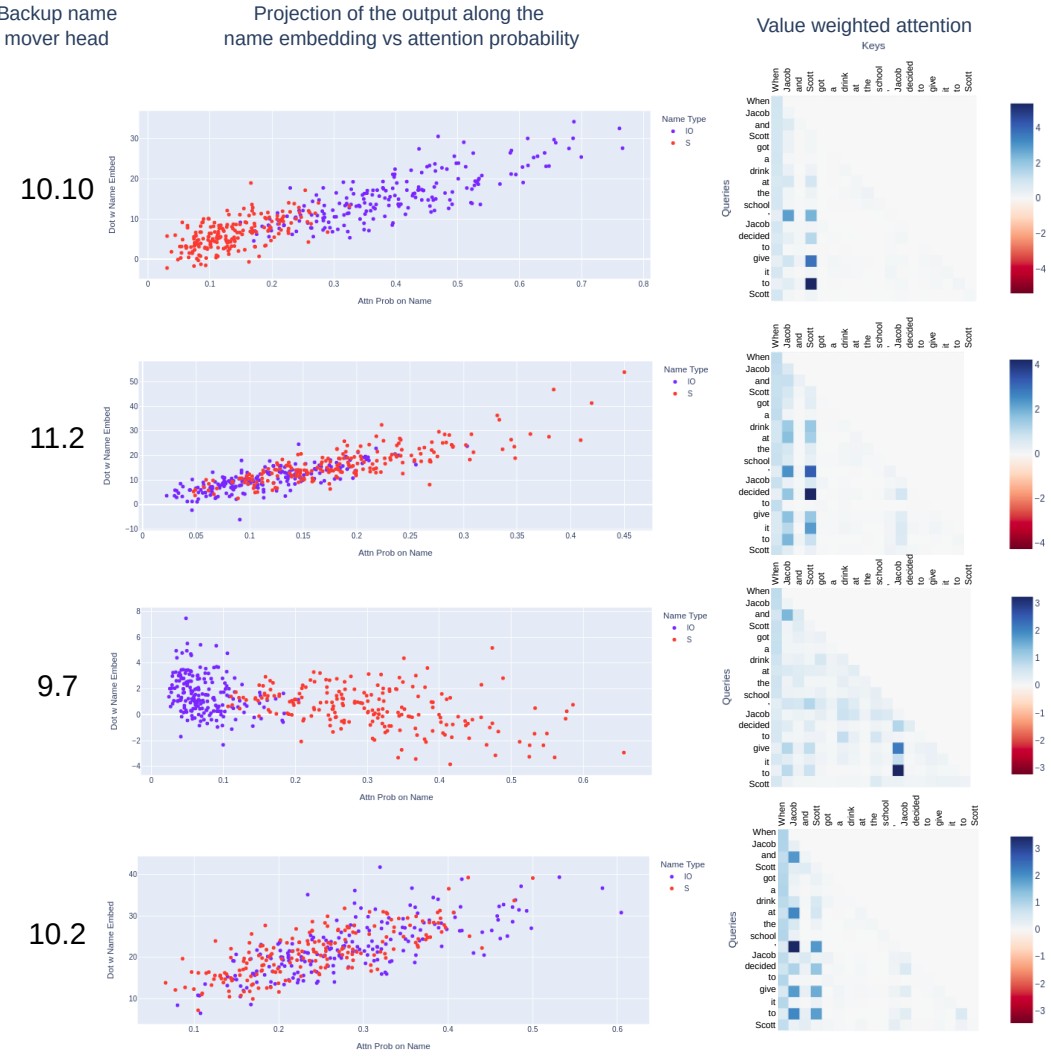

Figure 9: Four examples of Backup Name Mover Heads. Left: attention probability vs projection of the head output along $W_U[IO]$ or $W_U[S]$ respectively. Right: Attention pattern on a sample sequence.

| Distribution | Logit difference | IO probability | Proportion of S logit greater than IO |
|---|---|---|---|
| $p_{IOI}$ | 3.55 | 0.49 | 0.7% |
| Additional occurrence of S (natural sentence) | 3.64 | 0.59 | 0.4% |
| Additional occurrence of IO (natural sentence) | 1.23 | 0.36 | 23.4% |

Figure 10: Summary of GPT-2 performance metrics on the IOI task on different datasets. In the line order: for $p_{IOI}$, for the dataset where we added an occurrence of S (thus S appears three times in the sentence) and for the adversarial dataset with duplicated IO in natural sentences. IO probability refers to the probability the model places on the IO token (computed from the logits).

## C DESIGNING ADVERSARIAL EXAMPLES

As argued in Räuker et al. (2022), one way to evaluate the knowledge gained by interpretability work is to use it for downstream applications as predicting out of distribution behavior. In this section, we do this by using knowledge of the circuit to construct simple adversarial examples for the model.

As presented in Section 3, the model relies on duplicate detection to differentiate between S and IO. Motivated by this, we constructed passages where both the S and IO tokens are duplicated. An example is "John and Mary went to the store. Mary had a good day. John gave a bottle of milk to"; see Appendix D for full details. We find that this significantly reduces the logit difference and causes the model to predict S over IO 23% of the time (Figure 10).

To ensure that the observed effect is not an artifact of the additional sentences, we included a control dataset using the same templates, but where the middle sentence contains S instead of IO. In these sentences, S appears three times in total and IO only appears once. On this distribution, the model has an even higher logit difference than on $p_{IOI}$, and predicts S over IO only 0.4% of the time.

**Limitations of the attack.** Despite being inspired by our understanding of our circuit, those examples are simple enough that they could have been found without our circuit with enough effort.

Moreover, we do not have a full understanding of the mechanisms at play in these adversarial examples. For instance, the S-Inhibition Heads attend not only to S2, but also to the second occurrence of IO. As this pattern is not present in $p_{IOI}$ nor in ABC, it is beyond the analysis presented in Section 3. The study of the behavior of the circuit on these adversarial examples could be a promising area for future work.

## D TEMPLATE FOR ADVERSARIAL EXAMPLES

The design of adversarial examples relies on adding a duplicate IO to the sentences. To this end, we used a modification of the templates described in appendix A. We added an occurrence of [A] in the form of a natural sentence, independent of the context. The list of sentence is visible in Figure 11.

| |
|---|
| [A] had a good day. |
| [A] was enjoying the situation. |
| [A] was tired. |
| [A] enjoyed being with a friend. |
| [A] was an enthusiast person. |

Figure 11: Templates for the natural sentences used in the generation of adversarial examples. The sentences were chosen to be independent of the context.

## E GPT-2 SMALL FULL ARCHITECTURE

Here we define all components of the GPT-2 Architecture, including those we don't use in the main text. GPT-2 small has the following hyperparameters:

- $N$: number of input tokens.

- $V$: vocabulary of tokens.
- $d$: residual stream dimension.
- $L$: number of layers.
- $H$: number of heads per layer.
- $D$: hidden dimension of MLPs

It uses layer norms, the non-linear function

$$\text{LN}(x) \overset{\text{def}}{=} \frac{x - \bar{x}}{\sqrt{\sum_i (x_i - \bar{x}_i)^2}}, \tag{1}$$

where the mean and the difference from the mean sum are over the $d$ components of each of the $N$ tensors.

In GPT-2 the MLPs all have one hidden layer of dimension $D$ and use the GeLU non-linearity.

We addressed the parametrisation of each attention head in the main text, and cover the technical details of the $W_{QK}$ and $W_{OV}$ matrix here: the attention pattern is $A_{i,j} = \text{softmax}(x^T W_{QK}^{i,j} x)$ where the softmax is taken for each token position, and is unidirectional. We then have $h_{i,j}(x) \overset{\text{def}}{=} (A_{i,j} \otimes W_{OV}^{i,j}).x$.

---

**Algorithm 1** GPT-2.

---

**Require:** Input tokens $T$; returns logits for next token.
 1: $w \leftarrow$ One-hot embedding of T
 2: $x_0 \leftarrow W_E w$ (sum of token and position embeddings)
 3: **for** $i = 0$ to $L$ **do**
 4:      $y_i \leftarrow 0 \in \mathbb{R}^{N \times d}$
 5:      **for** $j = 0$ to $H$ **do**
 6:          $y_i \leftarrow y_i + h_{i,j}(x_i)$, the contribution of attention head $(i, j)$
 7:      **end for**
 8:      $y_i' \leftarrow m_i(x_i)$, the contribution of MLP at layer $i$
 9:      $x_{i+1} \leftarrow x_i + y_i + y_i'$ (update the residual stream)
10: **end for**
11: **return** $W_U \circ M \circ \text{LN} \circ x_L$

---

## F    ANALYSIS ON SEQUENCES OF RANDOM TOKENS

We run GPT-2 small on sequences of 100 tokens sampled uniformly at random from GPT-2's token vocabulary. Each sequence A was duplicated to form AA, a sequence twice as long where the first and second half are identical. On this dataset, we computed three scores from the attention patterns of the attention heads:

- The duplicate token score: for each token $T_i$ in the second half of a sequence $S$, we average the attention probability from $T_i$ to its previous occurrence in the first half of $S$ (i.e. $T_{i-100}$).

- The previous token score: we averaged the attention probability on the off-diagonal. This is the attention from the token at position $i$ to position $i - 1$.

- The induction score: the attention probability from $T_i$ to the token that comes after the first occurrence of $T_i$ (i.e. $T_{i-99}$)

These three score are depicted in Figure 12 for all attention heads. We can identify 3.0 and 0.1 as duplicated token heads that also appear in our circuit, 5.5 and 6.9 have high induction score and were also identified as induction heads in our investigation and 4.11 and 2.2 have a high previous token score. Note that the heads identified are also the ones that have the highest influence in the patching experiment shown in Figure 4.

**Induction Heads.** Olsson et al. (2022) define an Induction Head according to its behavior on repeated sequences of random tokens. The attention head must demonstrate two properties. i) Prefix-matching property. The head attends to [B] from the last [A] on pattern like [A] [B] ... [A] ii) Copy property. The head contribute positively to the logit of [B] on the pattern [A] [B] ... [A].

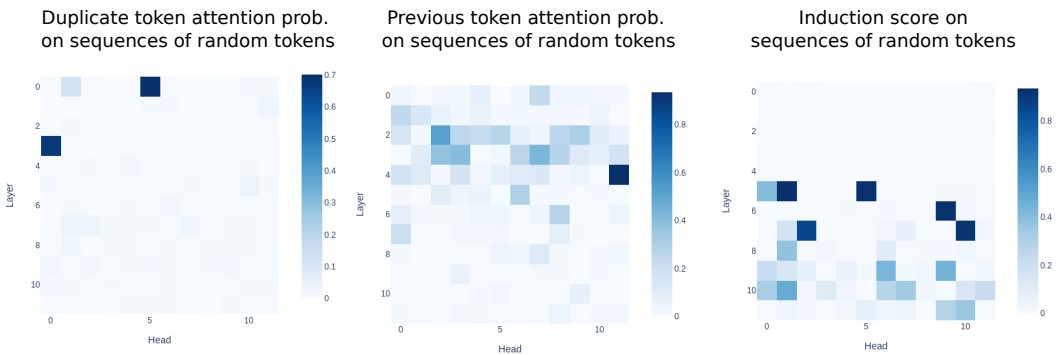

Figure 12: Sum of attention probabilities on position determined by the role. Left: duplicate score, the average attention probability from a token to its previous occurrence. Center: Previous token attention score, it is the average of the off diagonal attention probability. Right: Induction score. Average attention probability from the second occurence of [A] to [B] on [A][B]...[A].

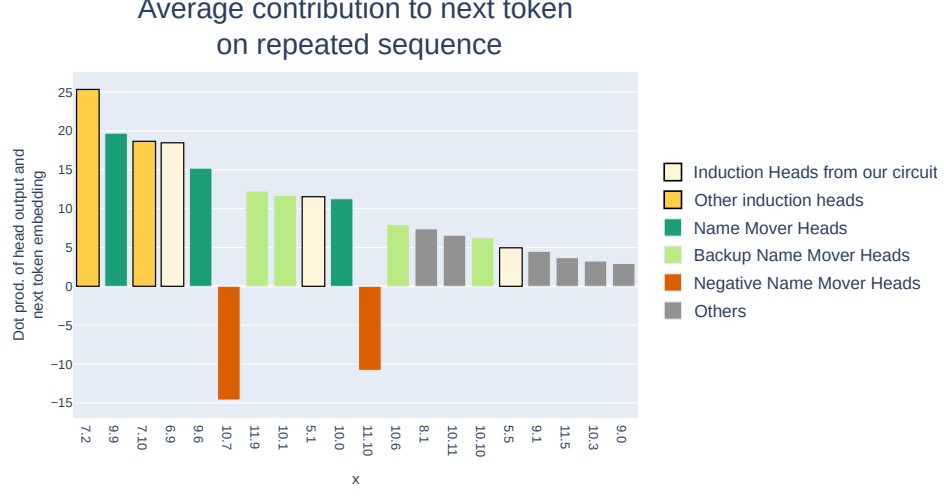

Figure 13: Contribution to the next token prediction per head on repeated sequences of tokens. The heads are ordered by decreasing absolute values of contribution. Black contour: heads with attention patterns demonstrating prefix matching property.

In the IOI task, we identify these heads according to their attention pattern, demonstrating the pattern-matching property. Here, we investigate their copy property, that is useless in the context of IOI: outputting the token after S2 is of no interest to identify IO.

As presented above, 5.5 and 6.9 are among the 5 heads with the highest induction score. This validates their prefix-matching property.

To check their copy property, we computed the dot product $\langle h_i(X), W_U[B] \rangle$ between the output of the head $h_i$ on sequence $X$ and the embedding of the token [B] on repeated sequences of random tokens. The results are shown in Figure 13. The two Induction Heads (5.5 and 6.9) appear in the 20 heads contributing the most to the next token prediction. Thus validating their copying property.

We also noticed that the majority of the Negative, Backup and regular Name Mover Heads appear to write in the next token direction on repeated sequences of random tokens, and Negative Name Movers Heads contribute negatively. This suggests that these heads are involved beyond the IOI task to produce next-token prediction relying on contextual information. Moreover, ablating the output of the three Name Mover Heads by patching their outputs results in a 26% increase in average loss on the last 99 tokens (from 0.15 to 0.19), showing their importance on tasks outside IOI.

# G  DISENTANGLING FEATURES IN THE OUTPUT OF S-INHIBITION HEADS

In Section 3.2, we discovered that S-Inhibition Heads are responsible for the Name Mover Heads' specific attention on the IO token. In this appendix, we explore which properties of the input affect the S-inhibition heads' outputs.

We present evidence that they were outputting *token signals* (information about the value of the token S), *positional signals* (related to the value of the position S1) and that the latter is the most important.

To disentangle the two effects, we design a series of counterfactual datasets where only some signals are present, and some are inverted with respect to the original dataset. We then conducted patching experiments where the output of S-Inhibition heads are computed from these datasets.

This enables us to quantify in isolation the impact of each signal on the final logit difference.

We constructed six datasets by combining three transformations of the original $p_{IOI}$ distribution.

- **Random name flip**: we replace the names from a given sentence with random names, but we keep the same position for all names. Moreover, each occurrence of a name in the original sentence is replaced by the same random name. When we patch outputs of S-Inhibition heads from this sentence, only positional signals are present, the token signals are unrelated to the names of the original sequence.
- **IO↔S1 flip**: we swap the position of IO and S1. The output of S-inhibition heads will contain correct token signals (the subject of the second clause is the same) but inverted positional signals (because the position of IO and S1 are swapped)
- **IO←S2 replacement**: we make IO become the subject of the sentence and S the indirect object. In this dataset, both token signals and positional signals are inverted.

We can also compose these transformations. For instance, we can create a dataset with no token signals and inverted positional signals by applying IO↔S1 flip on the dataset with random names. In total, we can create all six combinations of original, inverted, or uncorrelated token signal with the original and inverted positional signal.

From each of those six datasets, we patched the output of S-Inhibition heads and measured the logit difference. The results are presented in Figure 14.

These results can be summarized as the sum of the two effects. Suppose we define the variable $S_{tok}$ to be 1 if the token signal is the original, 0 when uncorrelated and -1 when inverted. And similarly $S_{pos}$ to be 1 if the position signal is the original and -1 if inverted. Then the Figure 14 suggests that the logit difference can be well approximated by $2.31 S_{pos} + 0.99 S_{tok}$, with a mean error of 7% relative to the baseline logit difference.

For instance, when both the positional and token signals are inverted, the logit difference is the opposite of the baseline. This means that the S token is predicted stronger than the IO token, as strong as IO before patching. In this situation, due to the contradictory information contained in the output of S-Inhibition heads, the Name Movers attend and copy the S1 token instead of the IO token (see Figure 15, right). In the intermediate cases where only one of the signals is modified, we observe a partial effect compared to the fully inverted case (e.g. Figure 15, left). The effect size depends on the altered signals: positional signals are more important than token signals.

Can we be more specific as to what the token and positional signals are? Unfortunately, we do not have a complete answer, but see this as one of the most interesting further directions of our work. We expect that the majority of the positional information is about the *relative* positional embedding between S1 and S2 (such pointer arithmetic behavior has already been observed in Olsson et al.

|  | **Original positional signal** | **Inverted position signal** |
|---|---|---|
| Original S token signal | 3.55 (baseline) | -0.99 |
| Random S token signal | 2.45 | -1.96 |
| S↔IO inverted token signal | 1.77 | -3.16 |

Figure 14: Logit difference after patching S-Inhibition heads from signal-specific datasets. The effect on logit difference can be decomposed as a sum of the effects of position and token signal.

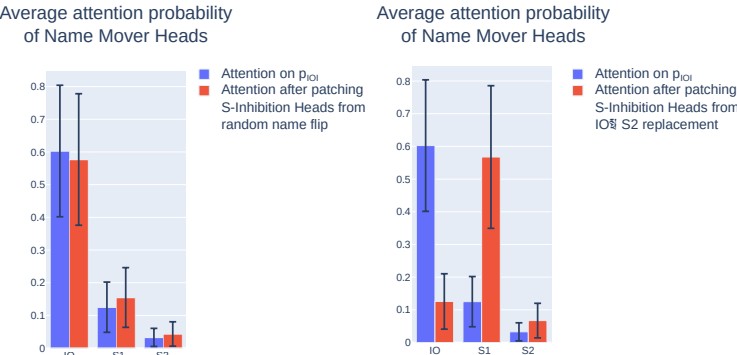

Figure 15: Name Mover Heads' attention probability before and after patching S-Inhibition Heads from signal-specific datasets. Left: patching from the dataset generated by random flip of name (same position signal, random token signal). Right: patching from the dataset generated by IO←S2 replacement (inverted position signal, inverted token signal). Black bars represent the standard deviation.

(2022)). When patching in S2 Inhibition outputs from a distribution where prefixes to sentences are longer (but the distance between S1 and S2 is constant), the logit difference doesn't change (3.56 before patching vs 3.57 after). This suggests that the positional signal doesn't depend on the absolute position of the tokens, as long as the relative position of S1 and S2 stays the same.

## H  LAYER NORM AND THE RESIDUAL STREAM

The attention heads and MLPs in GPT-2 small write into the residual stream. Suppose $x_{12}$ is the final state of the residual stream after the 12 layers. This is then converted into logits via $W_U \circ M \circ \mathrm{LN}(x_{12})$, where LN is defined in Appendix E, $M$ is the linear transformation of the layer norm operation and $W_U$ is the unembedding matrix.

In order to attribute the extent to which an attention head $h$ writes in a direction $W_U[T]$ where $T$ is a token (always IO or S in our case), we can't simply compute $\langle M \circ \mathrm{LN} \circ h_{i,j}(X), W_U[T] \rangle$, as the scaling factor that's used is $\sqrt{\sum_i (x_{12,i} - \overline{x_{12,i}})^2}$. Therefore $\overline{\mathrm{LN}}$ in the main text uses this scaling factor:

$$\overline{\mathrm{LN}}(h) \stackrel{\text{def}}{=} M \circ \frac{h - \overline{h}}{\sqrt{\sum_i (x_{12,i} - \overline{x_{12,i}})^2}} \tag{2}$$

## I  ROLE OF MLPS IN THE TASK

In the main text, we focused our investigation on attention heads. Since they are the only module able of moving information across token position – a crucial component of the IOI task – they were our main subject of interest. However, MLP can still play a significant role in structuring the residual stream at a given position. We explored this possibility by performing knock-out of the MLP layers (Figure 16). We observe that MLP0 has a significant influence on logit difference after knock-out ($-100\%$ relative variation) but the other layers don't seem to play a big role. We hypothesize that MLP0 can be used to perform low level token processing that latter layers rely on.

Moreover, we also investigated the writing of MLP along the $W_U[IO] - W_U[S]$ direction. As shown in Figure 16 (bottom) they write negligibly in this direction compared to attention heads (Figure 3).

| $v$ | Class | $K \cup \{v\}$ | $F(C \setminus (K \cup \{v\}))$ | $F(C \setminus K)$ |
|---|---|---|---|---|
| $(9, 9)$ | Name Mover | $[(9, 9)]$ | 2.78 | 3.14 |
| $(10, 0)$ | Name Mover | $[(9, 9), (10, 0)]$ | 2.43 | 2.78 |
| $(9, 6)$ | Name Mover | $[(9, 9), (10, 0), (9, 6)]$ | 2.77 | 2.43 |
| $(10, 7)$ | Negative Name Mover | All Negative Name Mover Heads | 5.11 | 3.84 |
| $(11, 10)$ | Negative Name Mover | All Negative Name Mover Heads | 5.11 | 4.06 |
| $(7, 3)$ | S-Inhibition | All S-Inhibition Heads | 0.33 | 1.15 |
| $(7, 9)$ | S-Inhibition | All S-Inhibition Heads | 0.33 | 1.12 |
| $(8, 6)$ | S-Inhibition | All S-Inhibition Heads | 0.33 | 1.10 |
| $(8, 10)$ | S-Inhibition | All S-Inhibition Heads | 0.33 | 0.55 |
| $(5, 5)$ | Induction | Induction Heads and Negative Heads | 1.06 | 3.95 |
| $(5, 8)$ | Induction | All Induction Heads | 1.06 | 2.58 |
| $(5, 9)$ | Induction | All Induction Heads | 4.40 | 5.11 |
| $(6, 9)$ | Induction | Induction Heads and Negative Heads | 4.76 | 5.11 |
| $(0, 1)$ | Duplicate Token | All Duplicate Token Heads | 1.14 | 2.52 |
| $(0, 10)$ | Duplicate Token | All Duplicate Token Heads | 1.14 | 2.29 |
| $(3, 0)$ | Duplicate Token | All Duplicate Token Heads | 1.14 | 1.65 |
| $(2, 2)$ | Previous Token | All Previous Token Heads | 2.03 | 2.80 |
| $(2, 9)$ | Previous Token | All Previous Token Heads | 2.03 | 2.42 |
| $(4, 11)$ | Previous Token | All Previous Token Heads | 2.03 | 2.27 |
| $(10, 10)$ | Backup Name Mover | All NMs and previous Backup NMs | 2.40 | 2.63 |
| $(10, 2)$ | Backup Name Mover | All NMs and previous Backup NMs | 0.89 | 1.09 |
| $(11, 2)$ | Backup Name Mover | All NMs and previous Backup NMs | 0.72 | 0.89 |
| $(10, 6)$ | Backup Name Mover | All NMs and previous Backup NMs | 2.63 | 2.77 |
| $(10, 1)$ | Backup Name Mover | All NMs and previous Backup NMs | 1.34 | 1.47 |
| $(9, 7)$ | Backup Name Mover | All NMs and previous Backup NMs | 0.85 | 1.02 |
| $(11, 9)$ | Backup Name Mover | All NMs and previous Backup NMs | 1.02 | 1.13 |
| $(11, 3)$ | Backup Name Mover | $[(9, 9), (10, 0), (9, 6), (10, 10), (11, 3)]$ | 2.53 | 2.59 |

Figure 17: $K$ sets for minimality for each $v$.

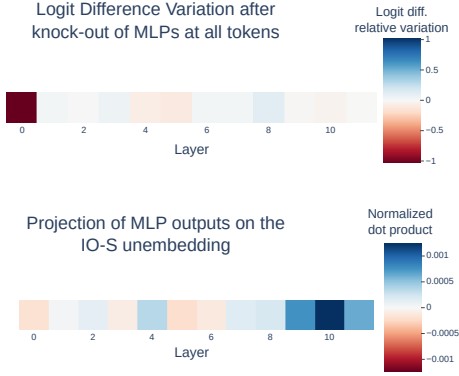

Figure 16: Top: Relative variation in logit difference from knocking out MLP layers. Only MLP0 causes a significative decrease in logit difference after knock-out. Bottom: We measure how much MLPs write along the $W_U[IO] - W_U[S]$ direction.

## J    MINIMALITY SETS

The sets that were found for the minimality tests are listed in Table 17.

| $K$ found by greedy optimization |
|:---:|
| (9, 9), (9, 6), (5, 8), (5, 5), (2, 2), (2, 9) |
| (9, 9), (11, 10), (10, 7), (8, 6), (5, 8), (4, 11) |
| (10, 7), (5, 5), (2, 2), (4, 11) |
| (9, 9), (11, 10), (10, 7), (11, 2), (3, 0), (5, 8), (2, 2) |

Figure 18: 4 sets $K$ found by the greedy optimization procedure on our circuit.

## K  GREEDY ALGORITHM

The Algorithm 2 describes the procedure used to sample sets for checking the completeness criteria using greedy optimization. In practice, because the naïve and the full circuit are not of the same size, we chose respectively $k = 5$ and $k = 10$ to ensure a similar amount of stochasticity in the process. We run the procedure 10 times and kept the 5 sets with the maximal important incompleteness score (including the intermediate $K$).

---
**Algorithm 2** The greedy sampling procedure for sets to validate the completeness citeria.

---
1: $K \leftarrow \emptyset$
2: **for** $i$ to $N$ **do**
3:      Sample a random subset $V \subseteq C$ of $k$ nodes uniformly.
4:      $v_{\text{MAX}} \leftarrow \arg\max_{v \in V} |\text{F}(C \setminus (K \cup \{v\})) - \text{F}(C \setminus K)|$
5:      $K \leftarrow K \cup \{v_{\text{MAX}}\}$
6: **end for**
7: **return** $K$

---

As visible in Table 18 the sets found by the greedy search contains a combination of nodes from different class. Nonetheless, the overlap between different $K$ suggest that we are missing components from $M$ that can take the place of induction heads or S-inhibition Heads when some Name Mover Heads are knocked-out.

## L  TECHNIQUES OVERVIEW

This work involved a variety of techniques that were required to explain model behavior.

- **Knockouts**:
  We used knockouts in two different ways: knocking out singular components of models, and knocking out everything in the model except particular circuits. The former was somewhat useful, and the latter we found powerful.
  - Knockout of single components: as an attribution method, knocking out singular components was not always as powerful as techniques such as projections, since the compensation (or backup) nature of Backup Name Mover Heads in this task allowed components to be knocked out and their true effect size masked.
  - Knockouts of all components except a circuit: on the other hand, knocking out all components except a circuit enabled us to isolate behaviors in this task where behavior was sparse, and check the components of our circuit while ignoring the vast percentage of components of the network, making work manageable.

  What was very important for the success of knockout and patching experiments was the choice of reference distribution for knockout. The analysis in Appendix G shows how the specific choice of dataset is useful for understanding model components. For a more general knockout, the OpenWebText dataset, GPT's training data, can be used. However, we found that this led to noisier results (though our circuit components still were shown to be important when we used this ablation).
- **Attention pattern analysis**:
  Using attention patterns to explain behavior is always worrying due to the possibility that information has accumulated on that token primarily from previous tokens, or that the position with large attention paid to isn't actually writing an important value into the residual stream. In our work however, analyzing attention patterns was generally a necessary first step before further experiments could be ran, and in this small model, both of the worrying cases did not generally arise.

- **Patching**:
  Patching was an important method we used to verify causal explanations that were generally formed from correlational evidence. In this way our use case is similar to Finlayson et al. (2021). We were surprised however that in general patching gave clear signal on the changes in behavior. This may be because we generally patched from inputs like the ABC distribution (which was successful in knocking out too). Therefore, keeping the context of the sentence templates may be generally useful. This could be either because the other words in the templates allow the model to realise that it should be doing IOI, or that introducing inputs from other distributions introduces noise that the model picks up on and uses, when this is not intended.

