# OpenReview forum: "Interpretability in the Wild: a Circuit for Indirect Object Identification in GPT-2 Small"
_ICLR.cc/2023/Conference — ICLR 2023 poster_

### Official Review · Reviewer_jy1a · 2022-10-20

**Confidence:** 4
**Correctness:** 2
**Technical Novelty And Significance:** 3
**Empirical Novelty And Significance:** 3
**Recommendation:** 3

**Clarity, Quality, Novelty And Reproducibility:**

Clarity:
- The writing was mostly clear, but the organization and style were not.  It lacks rigor in places.
- The work doesn't clearly outline its methodology and methodological assumptions.
- Poor organization: Section 3 doesn't clearly distinguish between experimental proceedures, results, and interpretation. Technical definitions and techniques are presented in a somewhat adhoc fashion throughout the paper.

Quality:
- Setting aside the issues with the work mentioned elsewhere in the review, the quality of the work is fairly high.  There are many interesting experiments and results.
- There are, however, insufficient ablation studies / sanity checks, and several of those performed question the validity of the interpretation (Section 3.4) or methods (Section 4.3).

Novelty:
- The novelty of the proposed criteria and techniques is not clearly stated.
- The paper lacks sufficient discussion of related work.  It doesn't discuss alternative approaches to interpretability, how their approaches and aims differ from mechanistic interpretability, or the pros/cons of mechanistic interpretability vs. other approaches.  It also doesn't make sufficiently clear claims about the novelty of this work vs. previous work on mechanistic interpretability.


Reproducibility:
- It wasn't clear how the model was trained.  Was it pretrained on natural language?  Fine-tuned on this task?  Trained only on this task?
- Identifying C seems to involve various judgment calls.  This is a fundamental barrier to reproducibility, which should not disqualify the work.  But the authors should be clear in identifying such judgment calls and stating the basis for making them.  How easily could the approach here be generalized to other tasks or models?  Can the authors provide a recipe that a reader could follow?  Or otherwise outline how this work could facilitate future work on mechanistic interpretability?

**Strength And Weaknesses:**


Strengths:
- The notions of faithfulness, completeness, and minimality seem like a promising approach to rigorously evaluating mechanistic interpretability claims.
- The paper covers a lot of ground; besides (apparently?) introducing these notions, they also propose methods for looking for circuits, and an interpretation of a particular model on a particular task.
- The interpretability efforts themselves are potentially useful as a case study that could guide or inform future work.
- The topic is neglected and potentially significant: mechanistic interpretability is worthy goal and not much has been published on it.

Weaknesses:
- The three claimed contributions are not sufficiently well supported.
1) RE "(1) We identify a large circuit in GPT-2 small that performs indirect-object identification (Figure 2 and Section 3)":
The discovery of the Backup Name Mover Heads suggests that other important bits of circuitry might've been missed, and that the model's behavior on this task might not be very well captured by the circuit.  Perhaps there are also Backup Backup Name Mover Heads, etc.?  Furthermore, as noted, computing the completeness score is intractable, the approximations produce somewhat inconsistent results, and no effort is made to determine which ones are reliable, excepting Section 4.3, where it is indirectly suggested both that the greedy method is both both better at estimating the completeness score, and that the
This is methodologically backwards: it assumes the interpretation is correct, and uses this as a basis to challenge the definition of the completeness score.
2) RE "(2) Through example, we identify useful techniques for understanding models, as well as surprising pitfalls":
This is a vague contribution and does not make a clear statement of novelty... which techniques are novel?  Or is it their application to "understanding models" that is novel?  What are the "surprising pitfalls"?
3) RE "(3) We present criteria that ensure structural correspondence between the circuit and the model, and check experimentally whether our
circuit meets this standard (Section 4)":
It has not been established that these criteria ensures structural correspondence; indeed, the notion of structural correspondence is not defined.  The criteria don't seem quite conceptually correct (see below).
The experimental results are not conclusive, and their interpretation remains subjective.  The checks that are performed seem ad-hoc and incomplete.
- There are a few limitations/concerns that deserve more discussion:
1) There seems to be an unstated assumption that circuits are they a meaningful way of understanding a model's behavior.  This assumption doesn't seem to be evaluated or supported by this work.
2) The paper seems optimistic about applying a similar approach to larger models, but doesn't discuss the challenges of scaling this approach (models may have a very high level of inherent complexity, mechanistic interpretability may require a prohibitive amount of human labor needed, even if models are simple enough to be understood mechanistically).
3) It seems the interpretation provided in this work can (at most) only tell us about how this model handles this very specific synthetic task.  How does this lead to a meaningful or practical understanding of how a model works more generally?  Or is the idea to study every behavior of interest in isolation?
- Overall, the writing is not careful in its claims (e.g. Section 3.2 starts with "Given that the Name Mover Heads are primarily responsible for constructing the output", but this hasn't been established).
- The definitions of faithfulness, completeness, and minimality do not seem fine-grained enough to fully capture mechanistic similarity between the circuit and the model.  I would expect the definition of F to measure the average difference in logits instead of the difference in average logits, and measure the difference in the vector of logits or probabilities instead of just the difference between IO and S tokens.  The work does not justify or discuss this chosen definition of F or alternatives.
- The paper should provide more motivation for this line of research.  What are the pros and cons versus other approaches to interpretability?  What is the value of understanding how the model behaves on this very specific set of inputs?  Can we draw more general conclusions about the model's behavior?  Or what is the end game?  What are the prospects for automating mechanistic interpretability?
- The definition of the task here is extremely narrow.  The model should be evaluated on OOD examples that similarly require outputting the correct name.  For instance, what if you add distractor text with other names in it to the context?  This work does not address how the model determines whether it should perform this task / engage this circuit.  Is there circuitry for recognizing what the task is?


Questions:
- Is the goal only to determine how the model solves this one task?  If some of the components of C played a completely different role in different tasks or on OOD examples, how much would that challenge the interpretation?
- The approach here seems based on finding attention heads that have large individual contributions, but couldn't there be a long-tail of other circuitry contributing to the model's behavior (e.g. in figure 6: what if you include more stuff?  why is .06 / 3% the magic cut-off)?  Section 3.4 suggests this might be the case.  Does this represent a fundamental challenge to the project of finding a circuit?  Consider an analogy with "typicality", where most of the probability mass of a distribution (such as a high-dimensional Gaussian) does not come from the most likely samples.
- For the "name mover heads", what does it mean that the OV matrix is a "name copying matrix"?  If they "copy whatever they attend to" (and not just names), is the claim that they are just approximating the identity matrix?  This could be investigated more.
- The role of the ABC distribution wasn't clear enough to me; can you clarify why it is used?

Minor comments:
- The names of the scores are counter-intuitive; e.g. the completeness score measures *in*completeness!
- The writing sometimes pretty unclear, e.g. "v can significantly recover F"; last 2 paragraphs of 3.3.
- "serial circuits" appears undefined.


**Summary Of The Paper:**

The paper finds a "circuit" (in this case, a subset of attention heads), C, that is meant to explain how GPT-2 small solves a particular synthetic task named Indirect Object Identification (IOI).
This task composes 15 templates of forms like "Then, [B] and [A] went to the [PLACE]. [B] gave a [OBJECT] to [A]".
The goal is to predict the name [A] at the end of the sentence, given all of rest of the sentence as context.
The model's behavior on other types of inputs was not evaluated.

The  claim is supported by experiments which aim to establish that C meets the authors' criteria of being faithful, complete, and minimal.
Some of their experimental results challenge this interpretation, however (see Sections 3.4 and 4.3).

The authors also describe how they go about identifying the proposed circuit, describing the techniques of knock-outs (i.e. looking at a subgraph of M), and patching (substituting different activations at some nodes of the computational graph while computing a forward pass).

The novelty of the proposed criteria and techniques is not clearly stated (except completeness, which they state they introduce).


**Summary Of The Review:**


The paper doesn't provide sufficient support for its claimed contributions.
In particular, I'm not convinced that the proposed circuit is an accurate representation of the model's behavior.
While it makes some meaningful progress in terms of techniques and methodology, more work is needed to justify and solidify these approaches; and the presentation also needs work.

---

> ### Author Response · Authors · 2022-11-15
> **Response to Reviewer jy1a [1/3]**
>
> We thank the reviewer for their criticisms of our work. We appreciate that the reviewer found our formulation of the criteria promising. We hope this comment will provide satisfactory answers to the questions the reviewer raised and clarify our contribution. We wish to draw the attention of the reviewer to the general comment where we address their concerns about the prospects of mechanistic interpretability, and our OOD examples.
>
> > 1. The discovery of the Backup Name Mover Heads suggests that other important bits of circuitry might've been missed, and that the model's behavior on this task might not be very well captured by the circuit. Perhaps there are also Backup Backup Name Mover Heads, etc.?
>
> If Backup Backup Name Mover Heads existed, this would mean that ablating the Backup and Regular Name Mover Heads would lead to a high incompleteness score, as the model would still be able to perform IOI much better than the circuit ablated in the same way. However, we observe this is not the case (Figure 5)---we find that the incompleteness score, $|F(M\setminus K)-F(C\setminus K)|$ when $K$ is the set of Regular and Backup Name Movers was 0.7 (<20% of the original logit difference).
>
> > 2. RE "(2) Through example, we identify useful techniques for understanding models, as well as surprising pitfalls": This is a vague contribution and does not make a clear statement of novelty... which techniques are novel? Or is it their application to "understanding models" that is novel?
>
> Most of the techniques used draw inspiration from previously existing techniques; we have edited the paper to clarify this.  The novelty is in their application to providing a fine-grained description of the largest circuit found in an "in the wild" language model to date. The techniques are:
>
> -   Unembedding projection has been used in the past to interpret the effect of full layers [(nostalgebraist, 2020)](https://www.lesswrong.com/posts/AcKRB8wDpdaN6v6ru/interpreting-gpt-the-logit-lens) and parameters [(Dar et al., 2022)](https://arxiv.org/abs/2209.02535). However, using unembedding projection to characterize the function of attention heads on a task is a novel use of this technique.
>
> -   The activation patching technique is similar to the causal mediation analysis introduced by [Vig et al., 2020](https://proceedings.neurips.cc/paper/2020/hash/92650b2e92217715fe312e6fa7b90d82-Abstract.html) and the causal tracing introduced in [Meng et al., 2022](https://openreview.net/forum?id=-h6WAS6eE4).
>
> -   Knockout draws from the wide literature of head pruning (e.g. [Michel et al., 2019](https://arxiv.org/abs/1905.10650)), and mean ablation was used before ([Nanda and Lieberum, 2022](https://www.alignmentforum.org/posts/N6WM6hs7RQMKDhYjB/a-mechanistic-interpretability-analysis-of-grokking)). However, the way we chose to compute the mean activations is specific to our work.
>
> The combination of the two first techniques in a strategy to systematically work backward from the logits to the input is also a contribution of this work. Also, all three of our proposed criteria are novel (we clarified this point in the introduction of Section 4).
>
> >  What are the "surprising pitfalls"?
>
> Our work characterizes surprising phenomena that are challenging for mechanistic interpretability:
>
> -   We identified several instances of heads implementing redundant behavior. The most surprising were "Backup Name-Mover Heads", which copy names to the correct position in the output, but only when regular Name-Mover Heads are ablated. This complicates the search for complete mechanisms, as different model structure is found when some components are ablated.
>
> -   We found known structures (specifically induction heads ([Elhage et al., 2021](https://transformer-circuits.pub/2021/framework/index.html))) that were used in unexpected ways. Thus mainline functionality of a component does not always give a full picture.
>
> -   Finally, we identified heads reliably writing in the opposite direction of the correct answer.
>
> We've edited the discussion to detail these points.
>
> > 1. There seems to be an unstated assumption that circuits are they a meaningful way of understanding a model's behavior. This assumption doesn't seem to be evaluated or supported by this work.
>
> We have added a citation to [Rauker et al., 2022](https://arxiv.org/abs/2207.13243) which describes "circuits analysis" (Section IV.D) as an approach to interpretability and how it has meaningfully described behavior in vision models and transformers. This assumption relies on a natural decomposition of the model architecture into simpler components (by attention heads and by token position), which are easier to understand individually. We can then combine the individual components to understand the behavior as a whole.

---

> ### Author Response · Authors · 2022-11-15
> **Response to Reviewer jy1a [2/3]**
>
> > it is indirectly suggested both that the greedy method is both both better at estimating the completeness score, and that the This is methodologically backwards: it assumes the interpretation is correct, and uses this as a basis to challenge the definition of the completeness score.
>
> We don't intend to challenge the definition of completeness; the failure to satisfy the adversarial completeness test is evidence that the circuit is not fully complete. Rather, our claims are that:
>
> -  The circuit is more complete than a more naive circuit (it passes the non-adversarial completeness test much better: see all points other than greedy points in Figure 6).
>
> -  The circuit is incomplete in a subtle rather than stark way--the failure of completeness was from subsets of heads from diverse classes (see Appendix K). Notably, the failure was not from the aforementioned "Backup Backup Name Movers", which would only be found when subsets of Name Movers were knocked out.
>
> > I would expect the definition of F to measure the average difference in logits instead of the difference in average logits
>
> > measure the difference in the vector of logits or probabilities instead of just the difference between IO and S tokens. The work does not justify or discuss this chosen definition of F or alternatives.
>
> The difference in logits is equal to the difference in log probabilities the model places on the two tokens, and hence the difference in loss the model would receive in training if IO was correct compared to if S was correct. The scope of the task we investigate is only the choice between these two tokens, we don't study e.g. alternative circuits that would be responsible to predict the token "the". Moreover, we chose to use difference and not distances as the sign of the difference matters: a logit difference of +3 and -3 should not be considered the same. And as the mean operation is linear, the average difference in logits and the difference in average logits are the same. We clarify this in section 2.
>
> **Questions**
>
> > Is the goal only to determine how the model solves this one task? If some of the components of C played a completely different role in different tasks or on OOD examples, how much would that challenge the interpretation?
>
> The goal of this work was to understand this particular task, as a first step toward mechanistic explanations of language models. As addressed in the general comment, we have found evidence that the lessons learnt from this work about components of C do transfer to different tasks. If some component of C played a completely different role in different tasks, this would indeed challenge our interpretation, as our mechanistic understanding of elements should still hold in other contexts.
>
> > The approach here seems based on finding attention heads that have large individual contributions, but couldn't there be a long-tail of other circuitry contributing to the model's behavior (e.g. in figure 6: what if you include more stuff? why is .06 / 3% the magic cut-off)? Section 3.4 suggests this might be the case. Does this represent a fundamental challenge to the project of finding a circuit?
>
> We think that this is an accurate description of a theoretical limitation of our method, but the fact that we manage to capture almost all model behavior with less than 2% of the parameters provides evidence against the long-tail of circuit components explaining most behavior. The cut-off we chose was a compromise between the circuit having large logit difference while keeping the number of nodes small enough such that the complete characterization of their end-to-end behavior was tractable. In our work, we found a compromise by qualitative consideration and future work could focus on finding a more principled approach to choose such cut-off.
>
> > For the "name mover heads", what does it mean that the OV matrix is a "name copying matrix"?
>
> As described in the "copy score" section (Section 3.1) the OV matrix transforms the residual stream of name tokens to a vector that when unembedded has large logits on the same name token. In effect, they are copying the token information from one position to another.
>
> We clarified this in the text.
>
> > The role of the ABC distribution wasn't clear enough to me; can you clarify why it is used?
>
> The ABC distribution allows us to define a baseline from which to find the model components that are specifically used for the IOI task. Sentences like "When Alice and Bob went to the store, Charlie gave a bottle of milk to" do not have a single indirect object that they should be completed with. However, these sentences have the same grammatical structure as the IOI sentences, and so when used for patching and ablation they don't introduce noisy information (e.g. the mean activation between a name token and a punctuation token).
>
> We clarified this section of the text.

---

> ### Author Response · Authors · 2022-11-15
> **Response to Reviewer jy1a [3/3]**
>
> **Minor Comments and  Clarity, Quality, Novelty And Reproducibility:**
>
> We thank the reviewer for the suggested edits, and we have addressed all minor comments in the main text.
>
> > It wasn't clear how the model was trained. Was it pretrained on natural language? Fine-tuned on this task? Trained only on this task?
>
> The model is GPT-2 introduced in [Radford et al., 2019](https://openai.com/blog/better-language-models/), a pretrained, and not fine-tuned model, available at <https://huggingface.co/gpt2>. We updated the paper to make this point clearer.
>
> > Can the authors provide a recipe that a reader could follow? Or otherwise outline how this work could facilitate future work on mechanistic interpretability?
>
> We have added a section to the appendix that provides an overview of the techniques we used, as well as takeaways to guide future readers wanting to reproduce our results (Appendix L). We also added a reference to this appendix from the main text.
>
> To facilitate replication, we will release the code after the anonymized discussion period.

---

### Official Review · Reviewer_wPfU · 2022-10-22

**Confidence:** 4
**Correctness:** 2
**Technical Novelty And Significance:** 2
**Empirical Novelty And Significance:** 2
**Recommendation:** 3

**Clarity, Quality, Novelty And Reproducibility:**

* Clarity: the writing is highly clear
* Novelty: the method is novel, as far as I know.
* Reproducibility: although code is not included, I believe that this paper and the method should not be hard to reproduce
* Impact: quite limited, which is my major concern as discussed in the weakness section.

**Strength And Weaknesses:**

Strengths:
1. Mechanistic interpretation is a new direction of interpretability research.
2. The analyzed model is GPT-2-small, of significant complexity.
3. The writing is clear and easily understandable.

Weaknesses:

Even though the paper tackles a very new area, for which I am fine with some rough and underdeveloped ideas, the general method suffers from too many flaws for me to "see the light at the end of the tunnel".

1. The method requires an _a priori_ known model working mechanism (e.g. the heuristic in Sec. 3), and basically maps different subgraphs (i.e. circuits) to various parts of this mechanism. This seems to be a very strong and possibly unjustified assumption: for all but the very synthetic tasks such as IOI, explicit rule-based classifiers (from the 80s and 90s) perform very poorly, suggesting that human-designed heuristics are most likely not how high-performing models work in general. Second, even if humans have a good grasp on one valid heuristic, it is likely that neural networks can use something totally different, due to accidental spurious correlations in the dataset, a problem known as shortcut learning [1].

2. The IOI task has three simple heuristic steps, mapping to the three major circuit classes. For more complex tasks, there may be tens or even hundreds of heuristics, especially for multi-class classification with a large number of classes (e.g. ImageNet). Identifying and interpreting them could be very cumbersome.

3. The introduction poses three advantages of mechanistic interpretation, "better predict out-of-distribution behavior, identify and fix model errors , and understand emergent behavior". However, none of them is demonstrated in the experiments, and it is not clear to me how they are enabled.

4. The paper assumes that attentions work "as expected", in that, e.g. "Name Mover Heads, by default, attend to previous names in the sentence, but due to the S-Inhibition Heads attend less to the S1 and S2 tokens". There have been numerous evidences suggesting that attentions may not be faithful interpretations [e.g. 2, 3, 4]. How would they be reconciled with the focus on attention in this paper?

[1] https://arxiv.org/abs/2004.07780

[2] https://arxiv.org/abs/1902.10186

[3] https://arxiv.org/abs/2004.03685

[4] https://arxiv.org/abs/2104.14403

**Summary Of The Paper:**

This paper provides a method for mechanistic interpretation of neural networks, specifically transformer models. The high-level idea is to derive a small subnetwork that can simulate the functionality of the full neural network while satisfying faithfulness, completeness, and minimality criteria. The paper empirically analyze the subnetwork using the GPT-2-small model trained on an indirect object identification (IOI) task. The proposed method is shown to identify several steps needed for the IOI task.

**Summary Of The Review:**

Overall, currently I am not convinced that mechanistic interpretation, as demonstrated in this paper, is a promising direction for understanding black-box models. Details are discussed in the weakness section. Therefore, I vote for rejection.

---

> ### Author Response · Authors · 2022-11-15
> **Response to Reviewer wPfU**
>
> We thank the reviewer for their comments and feedback. We appreciate that the reviewer found our work novel and our writing clear. We wish to draw the attention of the reviewer to the general comment where we address their concerns about the prospects of mechanistic interpretability and OOD examples.
>
> Before addressing the other comments, we would like to point out an unfortunate misconception about our paper:
>
> > The paper empirically analyze the subnetwork using the GPT-2-small model trained on an indirect object identification (IOI) task.
>
> As discussed in the introduction, the GPT-2 small model was not directly trained on the IOI task, we used the model from [Radford et al, 2019](https://openai.com/blog/better-language-models/), available at <https://huggingface.co/gpt2>. It is a language model trained on a large corpus of internet text data (Open Web Text). We found that this model was able to perform the IOI task without any fine-tuning. We updated the paper to make this point clearer.
>
> Below, we will address the reviewer's other comments point by point, including their impressions of the field of mechanistic interpretability and how our work fits into it.
>
> > 1. The method requires an a priori known model working mechanism (e.g. the heuristic in Sec. 3), and basically maps different subgraphs (i.e. circuits) to various parts of this mechanism.
>
> The working mechanism does not need to be known a priori for our approach to work.  We included the three-step algorithm in Section 3's front matter as an expository device to help guide the reader, not as a prerequisite of our methodology. The techniques in Sections 3.1 to 3.3 do not rely on a priori assumptions about the mechanism--instead, we trace information flow back from the logits to identify each of the components in turn. To clarify this point, we updated the introduction of the three-step algorithm in Sec. 3 to better convey the fact that it is a summary of the function performed by the circuit discovered after examining the model.
>
> What our method does rely on is that the computation is localized to a sparse set of attention heads. This may not be true for larger language models or other tasks and if so new techniques would need to be developed. But as long as we have this localization, we can find the components by starting at the end and working backward, without an a priori algorithm in mind.
>
> > This seems to be a very strong and possibly unjustified assumption: for all but the very synthetic tasks such as IOI, explicit rule-based classifiers (from the 80s and 90s) perform very poorly, suggesting that human-designed heuristics are most likely not how high-performing models work in general.
>
> The heuristic we presented is more flexible than the rule-based algorithms from the 80s/90s, because it can reference concepts (like "names") that would be hard to encode as a rule.
>
> > it is likely that neural networks can use something totally different, due to accidental spurious correlations in the dataset, a problem known as shortcut learning
>
> We agree that this is possible, and this is why we performed the completeness and faithfulness experiments that show that the neural network is not using additional heuristics. If the model were using a "shortcut" instead of the circuit we described, then our circuit alone could not perform the IOI task. However, we showed in section 4 that our circuit performed the task just as well as the full model (differing only by 6%, see "faithfulness score" in the text) and that the performance stays similar to the model under a wide range of knockouts (see the Sec. 4.1 on completeness).
>
> > 2. The IOI task has three simple heuristic steps...
>
> > 3. The introduction poses three advantages of mechanistic interpretation...
>
> We addressed these concerns in the general comment to all reviewers.
>
> > 4. The paper assumes that attentions work "as expected", in that, e.g. "Name Mover Heads, by default, attend to previous names in the sentence, but due to the S-Inhibition Heads attend less to the S1 and S2 tokens". There have been numerous evidences suggesting that attentions may not be faithful interpretations [e.g. 2, 3, 4]. How would they be reconciled with the focus on attention in this paper?
>
> We agree that attention patterns are not necessarily faithful interpretations of model behavior on their own, and mention this in Section 3.1: "since attention patterns can be misleading (Jain & Wallace, 2019), we check whether attention is correlated with the heads' functionality". Our experiments confirm our interpretations of the attention patterns. We show that attention is correlated with the head's effects on the name logits (Figure 3b) and also verify mechanistically that the Name Mover Heads copy names (Figure 3d).

---

### Official Review · Reviewer_sZ8X · 2022-10-24

**Confidence:** 3
**Correctness:** 4
**Technical Novelty And Significance:** 4
**Empirical Novelty And Significance:** 4
**Recommendation:** 8

**Clarity, Quality, Novelty And Reproducibility:**

Apart from what was discussed in the previous section, I have a question that the authors could perhaps clarify. I’m not too familiar with the literature, is the indirect object identification task an original contribution by the authors or is it from prior works? I didn’t see any reference but it was put under the background section, so I’m a bit confused. This task is a great task to study interpretability as authors rightly argued, as it is linguistically meaningful yet it is very clear what simple interpretable algorithm can perform well on this task.

Another potential improvement to clarity is adding a summary or overview of all methods used to identify circuit components in the introduction section, or somewhere else, so that reader can have a holistic idea about what is the complete set of techniques used to identify all the circuit components. Maybe expand a bit more on figure 1 in the introduction, I think that could be helpful in terms of readability. Without this overview, readers might need to put together this summary themselves as currently the organization of the circuit discovery is by circuit components rather than techniques used.

Nitpick: a typo in the last sentence of 4.3: “… worst-case completeness criteria” -> “… worst-case completeness criterion”


**Strength And Weaknesses:**

Strength.
Authors demonstrated interpretability techniques and criteria through example, making the abstract task of interpretability concrete.
Clarity: the paper is well written and easy to read.
Quality: The authors have formulated a set of three criteria to measure interpretability and have showed empirical evidence that their circuit have significant improvements compared to a naïve circuit.
Clear limitation: Authors are also being very clear on the limitations of their circuit, showing that their circuit does not do well with the completeness criterion.

Weakness
Failing the completeness criterion could be seen as a weakness but I tend to agree with the authors that worst-case completeness is too high a bar and could be explored in future works.


**Summary Of The Paper:**

In this work authors discovered specific transformer circuits for a natural language task called the indirect object identification, This task requires the LM to identify the indirect object and needs simple logical reasoning. The significance of this paper is that authors identified a large circuit in GPT-2 small that performs a relatively natural, linguistically meaningful task . Furthermore, the authors present a set of criteria that aim to measure the structural correspondence between the circuit and the model.

**Summary Of The Review:**

I’d recommend acceptance, as I think this paper is a solid contribution to the interpretability of language models in natural tasks that are linguistically meaningful.

---

> ### Author Response · Authors · 2022-11-15
> **Response to Reviewer sZ8X**
>
> We appreciate that the reviewer is optimistic about the impact of the work. In this comment, we respond to the question and suggested improvements.
>
> > I’m not too familiar with the literature, is the indirect object identification task an original contribution by the authors or is it from prior works?
>
> We introduced this task and to our knowledge, no prior work has studied it. We have updated the paper to make this point clearer.
>
> > Another potential improvement to clarity is adding a summary or overview of all methods used to identify circuit components in the introduction section, or somewhere else, so that reader can have a holistic idea about what is the complete set of techniques used to identify all the circuit components. Maybe expand a bit more on figure 1 in the introduction, I think that could be helpful in terms of readability. Without this overview, readers might need to put together this summary themselves as currently the organization of the circuit discovery is by circuit components rather than techniques used.
>
> We updated the paper to include a pointer to Appendix L from the main text. This section provides an overview of the techniques we used, as well as takeaways to guide future readers wanting to reproduce our results.
>
> > Nitpick: a typo in the last sentence of 4.3
>
> Thanks, we have updated the text accordingly.

---

### Official Review · Reviewer_iR9D · 2022-10-24

**Confidence:** 4
**Correctness:** 4
**Technical Novelty And Significance:** 3
**Empirical Novelty And Significance:** 4
**Recommendation:** 8

**Clarity, Quality, Novelty And Reproducibility:**

The paper is very well written, except section 4 which took me a second pass to fully grasp. The quality of the analysis is very high, hypotheses are derived sensibly and verified in multiple ways. As in many analysis case studies, there remains a certain amount of irreducible complexity, but e.g. the beginning of Section 3 is very helpful to set up the right intuitions. To the best of my knowledge the results are completely novel, and while some of the analysis techniques have been used before they have been combined in an exemplary fashion. Given the detailed description I think the analysis should be fairly easily reproducible, but I am not sure if all of the necessary details to do so are currently in the paper - maybe I missed if there is a planned code-release.

**Strength And Weaknesses:**

**Main contributions, Impact**
1) Well executed case study for identifying a mechanistic understanding of the circuit that solves the analysis task in GPt-2 small. Impact: medium - the techniques introduced to identify the circuit are not entirely novel, but are certainly put to good use; the identified circuit is interesting, but it is unclear whether the identified components will play a significant role in other analyses or a general understanding of strategies used by transformers to solve tasks.

2) Use of interventional analysis to establish causal relevance of identified circuit. This is crucial for any serious attempt at *explaining* network functionality; an explanation must lead to falsifiable (causal) predictions. While Knockouts are a standard technique for crude interventional analysis, the paper also uses more fine-grained Patching, demonstrating its usefulness for analysis of complex circuits. Impact: medium - falsification (or establishing causal relevance) is crucial for hypotheses established via interpretability/explainability techniques. While Knockouts are certainly not novel, and I am not sure whether Patching has been used in this precise context before (but would be somewhat surprised if nothing similar has been reported before), the consistent and effective use of the techniques in the paper serves as a good example for future analysis.

3) Analysis of the identified circuit in terms of faithfulness, completeness, and minimality. I particularly like that the paper aims at making these notions formal and quantitative, and reports quantitative results. For instance, without a check for completeness, the backup heads would likely not have been identified - which is an interesting result, highlighting the potential degrees of (complex) redundancies in neural circuitry. Impact: medium to high - the measures seem sensible, though perhaps not maximally universal, and I think that there is a good chance that they will be picked up by the community.

**Strengths**
 *  Well executed analysis case-study, showing that it is possible to open the “black box” or neural circuitry
 * Establishing of causal relevance of explanation, as well as quantitative definition and evaluation of faithfulness, completeness, and minimality of explanation
 * Serious attempts at verifying the correctness of the explanation, and “poking holes” to make sure the results are reliable.

**Weaknesses**
 * The biggest weakness to me is that the results are specific to a particular network (and as the paper suggest do not seem to easily transfer even to GPT-2 medium, a very closely related architecture) and task-instance (which allows for a fairly simple solution strategy as discuss at the beginning of Section 3). There is not much that can be done here - and I actually appreciate focusing on one meticulous analysis rather than analyzing a handful of tasks in a shallow fashion - but the limited generalizability of the results is one weakness of the work - I am more optimistic in terms of generalizing the methodology presented, though not all of it is entirely novel.
 * The second largest weakness to me is that the paper currently does not analyze whether the identified circuit has additional functionality (e.g. is used or even crucial for other tasks). Given the current writing one is inclined to (probably wrongly) infer that the (complete and minimal) sub-circuit found might exclusively serve to solve the particular task presented and that the sub-components also have clear and exclusive functionality (suggested by nomenclature such as ‘Name Mover heads’). I find it quite plausible that the sub-circuit and individual parts of it serve quite different functionality in different tasks, and are also partly involved in other (maybe even quite unrelated) tasks - e.g. there is no ‘name moving’ in an arithmetic task even though the same heads might potentially be involved. This exclusivity is not explicitly claimed in the paper, and perhaps not intended by the authors, but it would be great to see this discussed, and perhaps empirically analyzed (see improvements). Essentially, this is a word of caution regarding the recurrence of the well-known “polysemanticity” problem coined in C. Olah’s earlier work - perhaps the term “polyfunctionality” is more appropriate for circuits instead of individual neurons.

**Improvements**
1) Please discuss the possibility that the identified circuit and sub-parts (groups of heads) might be involved in other tasks where they might play a quite different functional role. Ideally (though this might be a stretch for a rebuttal phase) analyze whether knocking out the (minimal) circuit leads to performance losses in other, unrelated tasks. (I do not consider empirical results here necessary for publication).

2) Please discuss limitations of the current work slightly more prominently (the discussion already raises some points, but it would be great to see a strong and critical discussion of the limitations).

3) Please make sure that all details to reproduce the analysis are given in the appendix.

**Minor comments**

A) Though section 4 is fairly well written, it did take me two passes to grasp the quantitative metrics introduced for faithfulness, completeness, and minimality. I am not sure if pseudo-algorithms or graphical intuitions would be more helpful, but readability (or perhaps simply notation) could be improved a bit in Section 4. Not a major issue though.


**Summary Of The Paper:**

**Update after rebuttal** After reading the other reviews and the authors' responses I remain in favor of accepting the paper. I think wPfU, jy1a raise some very valid issues, but some of the issues raised concern the viability of the approach/research field of mechanistic interpretability in general, rather than the concrete results presented in the paper - which I think is valid to be discussed but to me is a less relevant criterion to judge one particular paper/analysis. jy1a raises additional relevant criticism specific to this particular paper, and in my opinion the authors have addressed that criticism sufficiently (I'd be keen to hear jy1a's opinion on this).

The paper is a case study analysis for identifying the mechanism / “algorithm” that GPT-2 small uses for solving a particular language task (Indirect object identification; indirect reference to one of two previously named persons in the same sentence to be precise). Backtracking from attention matrices - the paper essentially identifies groups of attention heads that each have an intuitive functionality, and, when combined, solve the task. To strengthen the identified circuit the paper defines metrics for faithfulness (circuit alone can explain good score on task), completeness (potentially duplicate parts of the circuit have been identified too), and minimality (no superfluous parts, that play no role in functionality and are not duplicates, have been identified) and confirms that the identified circuit fulfills all three notions. To demonstrate that the identified circuit is causally responsible for the functionality Knockouts (rendering parts of the network irrelevant) and Patching (splicing in activations from a different input into parts of the circuit, resembling a more fine-grained intervention than Knockouts) are used.

**Summary Of The Review:**

The paper is a strong example of an analysis case study to identify a simple algorithm for solving the indirect object identification task learned by GPT-2. Such case studies are rare, but luckily increasing in number - I personally think it is important to show the community to which degree it is actually possible to understand the internal (mechanistic) function of neural networks. I do not have any major concerns other than expanding the discussion around limitations and specificity of the findings to this particular task (same circuit or parts of it might be involved in other tasks, potentially even with different qualitative functionality). I currently think the work is polished and ready for publication and of interest to a fairly large audience.

---

> ### Author Response · Authors · 2022-11-15
> **Response to Reviewer iR9D**
>
>
> Thank you for the review of our work. We appreciate that you are optimistic about the relevance and impact of the paper. We recognize and agree with the limitations of our work (such as the open question of its scalability to different tasks and models), and address the specific comments and improvements here.
>
> > I find it quite plausible that the sub-circuit and individual parts of it serve quite different functionality in different tasks, and are also partly involved in other (maybe even quite unrelated) tasks
> > Please discuss the possibility that the identified circuit and sub-parts (groups of heads) might be involved in other tasks where they might play a quite different functional role. Ideally (though this might be a stretch for a rebuttal phase) analyze whether knocking out the (minimal) circuit leads to performance losses in other, unrelated tasks. (I do not consider empirical results here necessary for publication).
>
> We agree that the classes of heads we identified likely serve additional different functionality on other tasks, and in fact our analysis provides evidence for this: (1) induction heads are co-opted on our task in a non-standard way, and (2) back-up name-mover heads show multiple functionality. We have updated our work to provide further evidence of the induction heads’ behavior on different tasks in Appendix F.
>
> > Please discuss limitations of the current work slightly more prominently (the discussion already raises some points, but it would be great to see a strong and critical discussion of the limitations).
>
> We rewrote the discussion to more clearly state the limitations of our work.
>
> > Please make sure that all details to reproduce the analysis are given in the appendix.
>
> We have added additional details in Appendix L of the paper, and will release the code upon publication to make the analysis reproducible.
>
> > readability (or perhaps simply notation) could be improved a bit in Section 4
>
> We have edited Section 4 for clarity, including changing the “completeness score” to “incompleteness score”.

---

> > ### Comment · Reviewer_iR9D · 2022-11-29
> > **Thanks for the clarifications / comments and additional results**
> >
> > All my questions/concerns have been addressed; and I find it quite nice to see the additional results in Appendix F. Thanks!

---

### Author Response · Authors · 2022-11-15
**General response to reviewers**

We thank the reviewers for their time and careful feedback. We have addressed specific comments in replies to each reviewer. Here, we will respond to the general concerns that reviewers wPfU and jy1a have about the prospects of mechanistic interpretability. Also, we clarify the contribution of two appendices (C and F) in our updated submission on out-of-distribution (OOD) behavior that shows the application of our work beyond the case study we present in the main text.

**Prospects of Mechanistic Interpretability**

Reviewer wPfU is skeptical that mechanistic interpretability can scale to understanding large models and harder tasks, and reviewer jy1a asks for general motivation for work in this area. We argue below that i) mechanistic interpretability can plausibly scale to understanding a greater proportion of model behavior, and ii) even if we only ever achieve a mechanistic understanding of a small proportion of model behavior, the insights still may be valuable.

Firstly, mechanistic interpretability is a new field of study, and whether a comprehensive understanding of models can scale is an open question. Already, the field has explained behavior in models that use numerous heuristics. Prior works on ImageNet classifiers ([Olah et al., 2020](https://distill.pub/2020/circuits/curve-circuits/)), show that neural network components present regularities such that they can be studied in groups instead of individually, making their study less labor-intensive. This relates to our work, where we describe seven such groups. Given the increasing sophistication of the techniques available, we think that it is plausible that future studies could extend our method to cases involving circuits an order of magnitude larger. Descriptions of interactions involving hundreds of components are common in the fields of molecular biology (e.g. [metabolic pathways](https://en.wikipedia.org/wiki/Metabolic_pathway)) and neurology (e.g. [WormAtlas](https://www.wormatlas.org/)) which is evidence for the tractability of scaling this approach.

In addition, the phenomena discovered by mechanistic interpretability often provide broader insights than in the single circuit they were discovered in. For our circuit, Appendix F shows that the function of Previous Token, Duplicate Token, and Induction Heads extend to the case of repeated sequences of random tokens. Moreover, we extended the experiments presented in this appendix, and find that Name Movers and Negative Name Movers were also involved in this task. Additionally, Induction Heads ([Elhage et al., 2021](https://transformer-circuits.pub/2021/framework/index.html)) were found by prior mechanistic interpretability work and are repurposed for a different role in this task. Finally, prior work [(Nanda & Lieberum, 2022)](https://www.alignmentforum.org/posts/N6WM6hs7RQMKDhYjB/a-mechanistic-interpretability-analysis-of-grokking) studied how circuits form during the course of training to gain an understanding of the phenomenon of grokking.

**Out-of-distribution applications of our work**

To respond to the request (from reviewers wPfU and jy1a) for applications of our work beyond the IOI task defined in the main text, we have added a section on "adversarial out-of-distribution examples" (Appendix C) that exploit the fact that the model relies on duplicate detection to identify the IO. We created distractor sentences that repeated the indirect object to make the model incorrectly recognize IO as an improbable next token. For these adversarial examples, the model incorrectly predicts the subject (instead of the indirect object) 23.4% of the time, instead of 0.7% for IOI sequences. This shows that our understanding of the circuit enables us to predict out-of-distribution behavior (one of the three advantages of mechanistic explanation).

---

### Author Response · Authors · 2022-11-29
**Message to reviewers**

We would like to remind the reviewers that have only left initial reviews that we would appreciate having further feedback on our response. We are interested in hearing if our answers addressed all their concerns.

---

### Comment · Reviewer_jy1a · 2022-12-08
**Still in favor of rejection -- we should do a video chat.**

Apologies for the delay in responding.

I maintain that the paper is lacking precise claims and thorough evaluation of those claims.  Many of my concerns remain after the authors' response.  I believe more substantial changes would be needed for this paper to meet the standards of ICLR.  I understand that the revision period is over, but I believe the changes needed were already too large to address in a revision, and my opinion is that the paper should be substantially rewritten and experiments should be expanded before resubmitting.

I also don't believe the reviewers have properly addressed this concern of reviewer wPfU:
> The introduction poses three advantages of mechanistic interpretation, "better predict out-of-distribution behavior, identify and fix model errors , and understand emergent behavior". However, none of them is demonstrated in the experiments, and it is not clear to me how they are enabled.

The new OOD results are a step in the right direction, but I think much more would be needed to sufficiently support the authors claims:
- How does the model determine that an example belongs to the IOI "task"?  This seems like a critical part of explaining its behavior on these examples.  Does it perform this same behavior even when it shouldn't?  e.g. "When Maury and John knelt down to pray, John gave thanks to" should probably be completed with the name of some diety, not "Maury".  Does it?  What happens if there is additional context prepended to the example?  What happens if there is some other clauses inserted (that don't contain S or IO, or contain both of them some number of times, etc.)?
- I would also expect an analysis of cases where the model predicts something other than the IO (why does this happen?  Are these "errors" random, or if there some sense to them?)
- I believe there should be more work put into validating the methods used to evaluate faithfulness, completeness, and minimality.
- These are just three ideas; I think it is on the authors to devise and present a more convincing set of experiments.

On a related note, contra reviewer iR9D, the promise of mechanistic interpretability is quite important to discuss, since the paper claims in the abstract that: "Our work provides evidence that a mechanistic understanding of large ML models is feasible, opening opportunities to scale our understanding to both larger models and more complex tasks."  I am not convinced by this claim, and remain unconvinced by the central claims of the paper, which I think are overstated; in my mind, the evidence provided by this work is all still quite circumstancial.  I think this needs to be more clearly acknowledged in the writing, and also affects the framing of the contributions.

Given that we have substantial disagreement about the paper I suggest we have a video chat.


To the authors (RE response):
* Logit difference: my point was that you should care about the probabilities assigned to other tokens as well.  I also don't think the mean difference is a good measure, since errors in opposite directions should not cancel out.  If you want to say not only that C performs the IOI task to the same extent as M, but also *in the same way*, then you need to look at more fine-grained measures of behavior.

* Name mover heads: I'm still not clear on this point; do they copy whatever the attend to, or not?

---

> ### Author Response · Authors · 2022-12-13
> **Response to reviewer jy1a**
>
> Thanks for your detailed response! We would like to first apologize for a mistake in our definition of the IOI task that we think may have caused some confusion. We will then address the two questions individually.
>
> In the paper, we wrote ‘’The [IOI] task is to complete the main clause … with the non-repeated name (IO)”. We should have added that the task we study is completing the main clause **with IO rather than S**, as this is what the logit difference metric that we use measures. We will make this small addition in the camera-ready paper if our work is accepted.
>
> > Logit difference: my point was that you should care about the probabilities assigned to other tokens as well. I also don't think the mean difference is a good measure, since errors in opposite directions should not cancel out. If you want to say not only that C performs the IOI task to the same extent as M, but also in the same way, then you need to look at more fine-grained measures of behavior.
>
> Thanks for clarifying. We agree that using the mean difference (as opposed to the average absolute difference) in logit differences does mean that errors in opposite directions would “cancel out”. This is partially intentional: our goal in this project is to explain how the model does well on this task. In other words, we wanted to explain why the model does well on average; we did not want to have to explain the “noise” introduced by other parts of the model.
>
> That being said, we have reran many of our experiments with the mean absolute difference metric, and will add our results, as well as a discussion of why we chose to focus on the mean difference metric, to the appendix.
>
> Name mover heads: I'm still not clear on this point; do they copy whatever the attend to, or not?
>
> Yes, the name mover heads copy the IO and S whenever they attend to either the IO or S (Figure 3b). Note that we don't have any results for the Name Movers copying tokens other than the IO or S—it’s possible that they perform other roles on other tasks–since this behavior would not be relevant for the logit difference between IO and S.
>
> Finally, we’d also like to respond to this point:
> > I also don't believe the reviewers have properly addressed this concern of reviewer wPfU:
> > > The introduction poses three advantages of mechanistic interpretation, "better predict out-of-distribution behavior, identify and fix model errors , and understand emergent behavior". However, none of them is demonstrated in the experiments, and it is not clear to me how they are enabled.
>
> In the introduction, we cite past work that uses understanding of the mechanisms behind neural networks to perform each of the three applications. As for the viability of the specific circuits-based approach we use in this work, Olsson et al [1] and Nanda and Lieberum [2] demonstrate how recovering circuits inside of neural networks allow us to better understand their emergent behavior.
>
> While we do show how our understanding of the circuit allows us to predict out-of-distribution behavior, our primary contribution is not arguing for the specific use cases of (mechanistic) interpretability (which has been done in prior work).  Instead, our contribution is showing how circuits-based mechanistic interpretability can be scaled to non-toy language models and non-algorithmic tasks. As far as we know, this is by far the largest and most complicated circuit ever recovered on a language model. As we say in the abstract, our work “provide[s] evidence that a mechanistic understanding of large ML models is feasible”.
>
> [1] https://transformer-circuits.pub/2022/in-context-learning-and-induction-heads/index.html
>
> [2] https://www.alignmentforum.org/posts/N6WM6hs7RQMKDhYjB/a-mechanistic-interpretability-analysis-of-grokking

---

### Decision · Program_Chairs · 2023-01-20

**Decision:**

Accept: poster

**Justification For Why Not Higher Score:**

There are a few critical issues identified by reviewers:
1. Substantial changes needed for a stronger and critical limitation discussion.
2. There are a few not-well supported/overstated claims in the paper.

**Justification For Why Not Lower Score:**

Despite the fact that there are some critical issues raised by reviewers (also distinct scores), this paper is still an interesting and solid case study in the relatively new/small area – mechanistic interpretability. The methodology could be potentially interesting to the community to which degree it is possible to understand the internal function of neural networks.

**Metareview: Summary, Strengths And Weaknesses:**

Summary:
The paper presents an analysis for mechanisms that GPT-2 small uses for solving a particular language task, indirect object identification.

Strength:
1. The paper is a well executed analysis case-study in a relatively new/small mechanistic interpretability area .
2. The analysis of the formulated criteria (faithfulness, completeness, and minimality) in the identified circuit makes these notions formal and quantitative.
3. The methodology could be potentially interesting to the community to which degree it is possible to understand the internal function of neural networks.

Weakness:
1. As reviewers pointed out, some claims are overstated or not well supported.
2. Limitation of the work could be expanded to reflect all reviewers' concerns.
3. The analysis is only done for a particular network and a task-instance.

The major concerns from the reviewers are the current limited limitation section and a few not-well supported/overstated claims in the paper.
Request to the authors: Please update the paper to have a stronger and more critical limitation discussion, as well as substantially change the writing to justify all claims/assumptions (or not to overstate claims) in order to reflect reviewers’ comments.

**Note From Pc:**

if the above contains the word "oral" or "spotlight" please see: "oral" presentation means -> notable-top-5% and "spotlight" means -> notable-top-25%. As stated in our emails, we are disassociating presentation type from AC recommendations